# T-POP: TEST-TIME PERSONALIZATION WITH ONLINE PREFERENCE FEEDBACK

## ABSTRACT

Personalizing large language models (LLMs) to individual user preferences is a critical step beyond generating generically helpful responses. However, current personalization methods are ill-suited for new users, as they typically require either slow, resource-intensive fine-tuning or a substantial amount of pre-existing user data, creating a significant cold-start problem. To address this challenge, we introduce a new paradigm for real-time personalization by learning from online pairwise preference feedback collected during text generation. We propose T-POP (_Test-Time Personalization with Online Preference Feedback_), a novel algorithm that synergistically combines test-time alignment with _dueling bandits_. Without updating the LLM parameters, T-POP steers the decoding process of a frozen LLM by learning a reward function online that captures user preferences. By leveraging dueling bandits, T-POP intelligently queries the user to efficiently balance between exploring their preferences and exploiting the learned knowledge to generate personalized text. Extensive experiments demonstrate that T-POP achieves rapid and data-efficient personalization, significantly outperforming existing baselines and showing consistent improvement with more user interactions.

## 1 INTRODUCTION

While large language models (LLMs) have achieved remarkable success in generating human-like text, a critical frontier remains: moving from generic, one-size-fits-all responses to deeply personalized interactions. Users increasingly expect models to understand and adapt to their unique voice, style, and preferences (Zhang et al., 2024). The standard approach for aligning LLMs with human preferences has been through methods such as reinforcement learning from human feedback (RLHF) (Ouyang et al., 2022) and direct preference optimization (DPO) (Rafailov et al., 2023). However, these methods are primarily designed to align LLMs with _generic_ human preferences, failing to capture the specific nuances of individual users.

To address this gap, some recent works have adapted the RLHF framework to align LLMs with the preferences of individual users (Jang et al., 2023; Li et al., 2024b; Park et al., 2024; Lee et al., 2024). While effective, these approaches necessitate fine-tuning the LLM parameters for each user. Consequently, they are often unable to adapt quickly and efficiently to new users, posing a significant barrier to scalability and real-time personalization.

In response to the limitations of fine-tuning, another line of research has focused on personalization methods that do not require parameter updates. These techniques include retrieval-augmented generation (RAG) to fetch user-specific information (Sun et al., 2024; Mysore et al., 2023; Salemi et al., 2024) and the integration of the historical data of the user directly into the LLM prompt (Kang et al., 2023; Liu et al., 2023; Li et al., 2024a; Kim & Yang, 2024). A common prerequisite for these methods, however, is the availability of sufficient user data. This leaves them inapplicable to new users for whom such data has not yet been collected, a critical challenge in the field of personalization known as the _cold-start_ problem (Zhang et al., 2024).

To resolve this problem, a natural solution is to _collect user data online_ for new users. Drawing from the widespread success of RLHF and DPO, the most reliable and easily provided form of user data is _preference feedback_, where users indicate their relative preference between a pair of LLM-generated responses. We therefore propose to collect pairwise user preference data online to

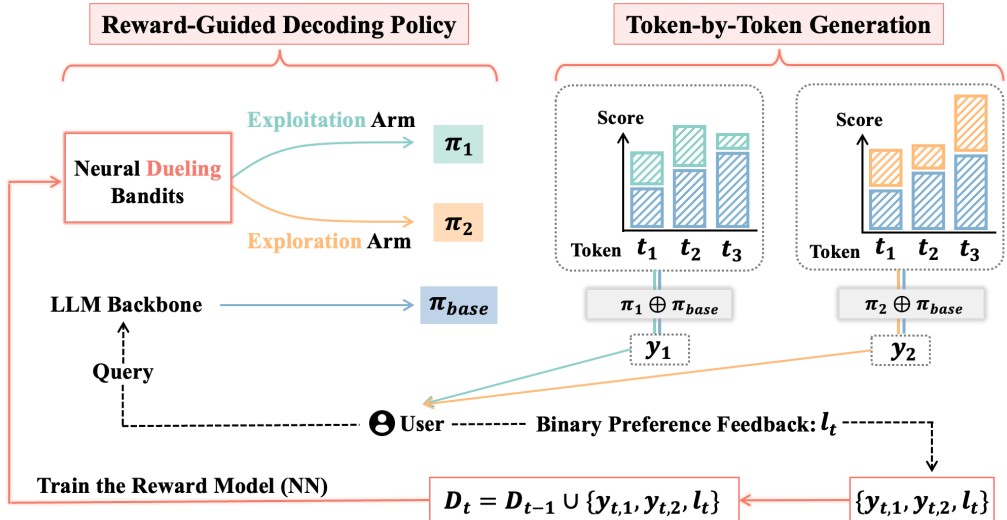

Figure 1: An overview of our `T-POP` for test-time personalization with online preference feedback.

facilitate rapid personalization. This approach, however, introduces a crucial challenge: *how do we simultaneously (1) collect user preference data online and (2) use these sequentially available data to achieve effective personalization?*

In this work, we tackle this challenge by proposing a principled combination of *test-time alignment* (Khanov et al., 2024) and *dueling bandits* (Verma et al., 2024). We introduce our Test-Time Personalization with Online Preference Feedback (`T-POP`) algorithm, which is illustrated in Fig. 1. Following the test-time alignment paradigm, `T-POP` adjusts the decoding process of a frozen LLM via an additive reward function that captures user personalization. This reward function is learned online and assigns higher values to responses that are better aligned with the personal preferences of the user. To learn this reward function effectively, we incorporate dueling bandits into the token selection process, which allows us to strategically select a pair of candidate tokens at every decoding step to query the user for feedback. Thanks to the inherent ability of dueling bandit algorithms to balance exploration and exploitation, our `T-POP` is able to simultaneously (1) generate high-reward responses that are increasingly aligned with user preference (i.e., exploitation) and (2) collect diverse preference data to rapidly refine the reward function (i.e., exploration). As a result, `T-POP` achieves effective user personalization using only a small number of online user feedback interactions.

In summary, our main contributions are:

- We formalize the problem of test-time personalization with online preference feedback, addressing the critical cold-start challenge for new users.
- We propose `T-POP`, a novel algorithm that synergistically combines test-time alignment with dueling bandits to achieve rapid, data-efficient personalization without any parameter fine-tuning.
- Extensive experiments show that `T-POP` significantly outperforms existing personalization baselines, with its effectiveness steadily increasing as more user feedback is provided.

## 2 PRELIMINARY

**Test-Time Alignment for Personalization.** Our work builds upon the paradigm of *test-time alignment*, which steers the generation process of a frozen LLM at inference time without updating its parameters. The core idea is to guide each token selection step towards user-preferred outcomes. Specifically, given a partially generated sequence $y_{<p}$, the standard approach is to sample the next token $y_p$ from the probability distribution of the base LLM $\pi_{\text{base}}(\cdot|y_{<p})$. To incorporate personalization, we introduce a reward function $r(\cdot; \theta)$ parameterized by $\theta$, which is learned online to capture the user preferences. This reward function assigns a scalar score to any given sequence, with *higher scores indicating better alignment with the preference of the user*.

At each decoding step $p$, we define a scoring function that combines the base model's likelihood with the learned preference reward. For any candidate token $v$ from the vocabulary $\mathcal{V}$, the score is calculated as:

$$\text{Score}(v|y_{<p}) = \pi_{\text{base}}(v|y_{<p}) + \omega \cdot r([y_{<p}, v]; \theta) \tag{1}$$

where $[y_{<p}, v]$ denotes the new sequence formed by appending token $v$ to the prefix $y_{<p}$, and $\omega$ is a hyperparameter controlling the strength of the personalization. The decoding policy then selects the next token by maximizing this score: $y_p = \arg\max_{v \in \mathcal{V}} \text{Score}(v|y_{<p})$. This framework allows the generation to be dynamically steered towards personalized content by optimizing a local, per-token objective. In our problem of test-time personalization without sufficient user data, the central challenge is how to learn the reward function $r(\cdot; \theta)$ efficiently from online preference feedback from the user. To this end, we adopt the framework of neural dueling bandits.

**Neural Dueling Bandits.** To learn the reward function $r$ from online preference feedback, we frame the problem within the neural dueling bandits framework (Verma et al., 2024). This setting is designed for learning from pairwise preference feedback (e.g., "response A is better than response B"), which is often more reliable and easier for users to provide than absolute scores.

In this framework, a learner iteratively interacts with a user. In each round, it presents a pair of items (i.e., arms), and the user provides feedback indicating which one they prefer. The user's choice is assumed to be governed by the underlying reward function $r$. This relationship is commonly modeled using the Bradley-Terry-Luce (BTL) model (Hunter, 2004; Luce et al., 1959), which states that the probability of preferring arm $a_1$ over arm $a_2$ is given by: $P(a_1 \succ a_2) = \sigma(f(a_1) - f(a_2))$, where $f$ denotes the unknown reward function and $\sigma(z) = 1/(1 + e^{-z})$ is the sigmoid function. To learn complex user preferences in text generation, we adopt a neural network (NN) $r(\cdot; \theta)$ parameterized by $\theta$ to approximate $f$ (Verma et al., 2024).

## 3 THE T-POP ALGORITHM

In this section, we introduce our T-POP algorithm (Fig. 1, Algo. 1), which addresses the cold-start personalization problem for new users. We begin by discussing the high-level insights behind our approach, followed by a detailed breakdown of its components.

### 3.1 HIGH-LEVEL OVERVIEW

The core insight behind T-POP is the synergistic integration of test-time alignment with the principles of online learning from dueling bandits. Instead of treating personalized text generation and user preference learning as separate phases, T-POP interweaves them into a single, efficient process. The algorithm operates by steering the decoding of a frozen LLM to simultaneously generate two competing sequences in real-time.

This is achieved by applying a dueling bandit policy at *each token-generation step*. The **exploitation sequence** is constructed by greedily following the reward model's current estimate of user preferences (line 10 of Algo. 1). Concurrently, the **exploration sequence** is built by optimistically choosing tokens that balance high estimated reward with high uncertainty (line 11 of Algo. 1). The two completed responses are then presented to the user, who provides feedback on which one they prefer. This feedback is immediately used to update the reward model, improving its alignment with the user preferences. This creates a tight feedback loop: the dueling bandit policy generates **personalized and informative pairs of responses** for learning, and the user feedback immediately refines the reward model, which in turn improves the personalized text-generation policy for the next round of interaction. This entire process requires no fine-tuning of the base LLM, enabling rapid and data-efficient personalization with online feedback.

### 3.2 ONLINE PERSONALIZATION LOOP

T-POP operates over a series of interaction rounds $t = 1, 2, \ldots, T$. The goal in each round is to generate a personalized and informative pair of responses $(y_{t,1}, y_{t,2})$, elicit user preference feedback $l_t$, and update the neural network reward model $r(\cdot; \theta)$.

The learning process begins with an initial reward model $r(\cdot; \theta_1)$. In each round $t$, the algorithm generates the pair $(y_{t,1}, y_{t,2})$ based on the current reward model $r(\cdot; \theta_t)$, as detailed in Sec. 3.3.

---

**Algorithm 1:** `T-POP`

---

**Input:** Initial reward model parameters $\theta_1$, matrix $V_0 = \lambda I$, number of user interactions $T$, reward weight $\omega$, exploration parameter $\nu$, number of candidate tokens $k$, maximum number of tokens $M$ in a response, observation history $\mathcal{D}_0 = \mathcal{I}$.

1 **for** $t = 1, \ldots, T$ **do**

2    Receive the user query $q_t$ in the current round, set $y_{t,1} = [q_t]$, $y_{t,2} = [q_t]$

3    **for** *each token position* $p = 1, \ldots, M$ **do**

4      $\mathcal{V}_p^{(1)} \leftarrow$ top-$k$ tokens conditioned on $y_{t,1}$

5      $\mathcal{V}_p^{(2)} \leftarrow$ top-$k$ tokens conditioned on $y_{t,2}$

6      $\mathcal{V}_p \leftarrow \mathcal{V}_p^{(1)} \cup \mathcal{V}_p^{(2)}$

7      **for** $v \in \mathcal{V}_p$ **do**

8        $score_1(v; \theta_t) \leftarrow \pi_{\text{base}}(v|y_{t,1}) + \omega \cdot r\left([y_{t,1}, v]; \theta_t\right)$

9        $score_2(v; \theta_t) \leftarrow \pi_{\text{base}}(v|y_{t,2}) + \omega \cdot r\left([y_{t,2}, v]; \theta_t\right)$

10      Select token for response 1: $v_{p,1} \leftarrow \arg\max_{v \in \mathcal{V}_p} score_1(v; \theta_t)$

11      Select token for response 2:
       $v_{p,2} \leftarrow \arg\max_{v \in \mathcal{V}_p} score_2(v; \theta_t) + \omega \cdot \nu \left\| \nabla r([y_{t,2}, v]; \theta_t) - \nabla r([y_{t,1}, v_{p,1}]; \theta_t) \right\|_{V_{t-1}^{-1}}$

12      $y_{t,1} \leftarrow [y_{t,1}, v_{p,1}], y_{t,2} \leftarrow [y_{t,2}, v_{p,2}]$

13      $V_{t-1} \leftarrow V_{t-1} + (\nabla r(y_{t,1}; \theta_t) - \nabla r(y_{t,2}; \theta_t))(\nabla r(y_{t,1}; \theta_t) - \nabla r(y_{t,2}; \theta_t))^\top$

14    Obtain binary user preference feedback $l_t = \mathbb{1}_{\{y_{t,1} \succ y_{t,2}\}}$ and update history:
     $\mathcal{D}_t = \mathcal{D}_{t-1} \cup (y_{t,1}, y_{t,2}, l_t)$;

15    Train NN using history $\mathcal{D}_t = \{(y_{s,1}, y_{s,2}, l_s)\}_{s=1,\ldots,t}$ by minimizing loss function $\mathcal{L}_t(\theta)$
     (equation 2): $\theta_{t+1} = \arg\min_\theta \mathcal{L}_t(\theta)$

16    Update the covariance matrix: $V_t \leftarrow V_{t-1}$

---

The user then provides a binary preference $l_t = \mathbb{1}_{\{y_{t,1} \succ y_{t,2}\}}$, which is equal to 1 if the response $y_{t,1}$ is preferred over $y_{t,2}$ and 0 otherwise. This new data point is then added to the history $\mathcal{D}_t = \mathcal{D}_{t-1} \cup \{(y_{t,1}, y_{t,2}, l_t)\}$ (line 14 of Algo. 1). Upon receiving this feedback, the parameters of the reward model (i.e., neural network) are updated by minimizing the following loss function over the entire history $\mathcal{D}_t$ (line 15 of Algo. 1):

$$\mathcal{L}_t(\theta) = - \sum_{(y_1, y_2, l) \in \mathcal{D}_t} \left[ l \log \sigma(r(y_1; \theta) - r(y_2; \theta)) + (1-l) \log \sigma(r(y_2; \theta) - r(y_1; \theta)) \right] + \lambda \|\theta\|_2^2, \quad (2)$$

in which $\sigma(\cdot)$ is the sigmoid function. Of note, minimizing this loss function (equation 2) is equivalent to *maximizing the log-likelihood of the preference observations* $\mathcal{D}_t$ according to the Bradley-Terry-Luce (BTL) model (Sec. 2), plus a regularization term (Verma et al., 2024). This updated reward model, with parameters $\theta_{t+1} = \arg\min_\theta \mathcal{L}_t(\theta)$, is then used in the next round, enabling continuous improvement of the reward model from user interactions.

**Continuous Deployment via Asynchronous Learning.** Contrary to a rigid "collect-then-deploy" paradigm, `T-POP` is designed for continuous, low-latency deployment throughout the interaction. By decoupling model updates from user interactions, `T-POP` can minimize latency increase:

- **Asynchronous Online Updates:** To eliminate the training latency, we implement an asynchronous update strategy. When a user provides preference feedback at round $t$, the reward model update ($\theta_t \rightarrow \theta_{t+1}$) is triggered in a background thread. Crucially, during the model update process, our `T-POP` continues to serve subsequent queries *using the latest reward model* $r(\cdot; \theta_t)$. After the model update concludes, the updated reward model $r(\cdot; \theta_t)$ will then be used to serve subsequent user queries. This ensures that the computational cost of training is completely masked from the user experience.
- **Flexible Deployment Mode:** Once the personalization phase concludes (at any arbitrary interaction $t$), `T-POP` transitions to a definitive inference mode. The learned reward model, $r(\cdot; \theta_t)$, is frozen and utilized solely by the exploitation arm. Generation then proceeds via *token-by-token greedy decoding*, where each token is selected to maximize the score in equation 1 based on the final reward model. This effectively crystallizes the learned preferences into a standard, low-overhead text generator.

## 3.3 TOKEN-BY-TOKEN ARM GENERATION

A key innovation of `T-POP` is its dynamic, token-by-token construction of the dueling sequences, $y_{t,1}$ and $y_{t,2}$, which is achieved by integrating dueling bandits with reward-guided decoding. The pair of sequences is built over $M$ steps (lines 3–13 of Algo. 1), with the exploitation-exploration policy applied at each step to select the next token for each growing sequence.

**Exploitation Sequence.** The first sequence, $y_{t,1}$, represents pure *exploitation*. It is generated to be the best possible response according to the current reward model $r(\cdot; \theta_t)$. At each token position $p$, the next token $v_{p,1}$ is chosen greedily to maximize the reward-guided scoring function from equation 1:

$$v_{p,1} = \underset{v \in \mathcal{V}_p}{\operatorname{argmax}} \left( \pi_{\text{base}}(v|y_{t,1}) + \omega \cdot r([y_{t,1}, v]; \theta_t) \right), \tag{3}$$

where $\mathcal{V}_p$ is a set of candidate tokens formed by the top-$k$ tokens from the base LLM (Algo. 1, lines 4-6). This process iteratively builds a sequence aligned with the current reward model $r(\cdot; \theta_t)$.

**Exploration Sequence.** The second sequence, $y_{t,2}$, simultaneously accounts for exploitation and *exploration*. That is, it aims to not only achieve high reward values to align with the user preference (i.e., exploitation), but also generate informative responses with *large uncertainty* to accelerate the learning of the reward model (i.e., exploration). Specifically, at each token position $p$, it selects the next token $v_{p,2}$ by maximizing the sum of the score and a UCB-style exploration bonus:

$$v_{p,2} = \underset{v \in \mathcal{V}_p}{\operatorname{argmax}} \underbrace{\pi_{\text{base}}(v|y_{t,2}) + \omega \cdot r([y_{t,2}, v]; \theta_t)}_{\text{Exploitation}} + \underbrace{\omega \cdot \nu \cdot \text{UncertaintyBonus}(v)}_{\text{Exploration}}. \tag{4}$$

The uncertainty bonus term is defined as:

$$\text{UncertaintyBonus}(v) = \|\nabla r([y_{t,2}, v]; \theta_t) - \nabla r([y_{t,1}, v_{p,1}]; \theta_t)\|_{V_{t-1}^{-1}}. \tag{5}$$

Our generation strategy is grounded in the theoretically principled Neural Dueling Bandit framework (Verma et al., 2024) and the Tokenized Bandit theory (Shin et al., 2025).

**Guarantees for Neural Dueling Bandits.** The matrix $V_{t-1}$ (line 14 of Algo. 1) aggregates the gradient information from all previously selected sequences:

$$V_{t-1} \leftarrow V_{t-1} + (\nabla r(y_{t,1}; \theta_t) - \nabla r(y_{t,2}; \theta_t))(\nabla r(y_{t,1}; \theta_t) - \nabla r(y_{t,2}; \theta_t))^{\top} \tag{6}$$

This covariance update allows the uncertainty bonus in equation 5 to measure the epistemic uncertainty of a candidate sequence $[y_{t,2}, v]$ relative to the exploitation arm $[y_{t,1}, v_{p,1}]$. As established by Verma et al. (2024), maximizing this gradient-based bonus ensures that the system efficiently explores the reward parameter space. Under standard regularity assumptions (e.g., bounded norm in a Reproducing Kernel Hilbert Space), this mechanism achieves a cumulative regret bound of $R_T = \tilde{O}(d_{eff}\sqrt{T})$, where $d_{eff}$ is the effective dimension of the neural tangent kernel matrix. This theoretical result guarantees that our reward model converges to the user's true preference with high probability.

**Guarantees for Sequential Decoding.** Extending bandit guarantees to token-by-token generation is non-trivial due to the combinatorial search space. However, our approach is supported by the recent findings of Shin et al. (2025), who proved that linear bandit algorithms applied to token-level decoding achieve sublinear regret $R_T = \tilde{O}(L\sqrt{T})$, provided the utility function satisfies the *Diminishing Distance with More Commons (DDMC)* assumption. Here $L$ denotes the maximum sequence length. Therefore, `T-POP` effectively operationalizes these theoretical principles: the uncertainty bonus steers generation towards sequences that provide significant novel information (exploration), while the reward score ensures alignment (exploitation), theoretically ensuring both sample efficiency and convergence in the sequential decoding setting.

## 4 EXPERIMENTS

We conduct comprehensive experiments to empirically validate the effectiveness and data efficiency of our `T-POP`, particularly its ability to achieve rapid personalization in cold-start scenarios. Some experimental details are deferred to App. B due to space constraints.

## 4.1 Experimental Setting

**Models, Datasets and Personalization Attributes.** We conduct experiments on a diverse set of modern open-source LLMs, including Mistral-7B-Instruct-v0.2 (Jiang et al., 2023), Llama-3.1-8B-Instruct (Grattafiori et al., 2024), and Qwen2-7B-Instruct (Yang et al., 2025). Our evaluation suite is built upon four established benchmarks to ensure a comprehensive assessment. We use (1) **HelpSteer** (Wang et al., 2023) for its multi-faceted instruction-following challenges and two subsets of **UltraFeedback** (Cui et al., 2024): (2) **TruthfulQA** (Lin et al., 2021) and (3) **UltraChat**—to evaluate factuality and conversational ability, respectively. To directly measure alignment with user tastes, we also include the (4) **Personal Preference Eval** (Gao et al., 2024) dataset. To simulate diverse real-world user preferences, we evaluate our method across four distinct preference attributes, inspired by prior work (Zhong et al., 2024; Zhang et al., 2025b): *creative*, *verbose*, *concise*, and *uplifting*.

**Baseline Methods.** We compare our T-POP against a suite of strong baselines representing different personalization paradigms. These include the original, unmodified backbone LLM (**Base**); the backbone guided only by prompt engineering (**Preference Prompting (Pref)**); and a standard decoding algorithm, **Beam Search (BS16)**, with a beam width of 16. We also compare against two state-of-the-art training-free methods: **Linear Alignment (LA)** (Gao et al., 2024), which linearly updates the model's logits to steer generation, and our primary competitor, **AMULET** (Zhang et al., 2025b), which formulates token-level decoding as an online learning problem for test-time alignment.

**Evaluation Metrics.** Given the subjective nature of personalization, we employ a two-pronged evaluation strategy. Our primary quantitative metric is the **Reward Model Score**. We use the widely used **ArmoRM-Llama3-8B-v0.1** (Wang et al., 2024) to score the alignment of generated responses with the target attribute, following the evaluation methodology of Zhang et al. (2025b). To complement this and capture nuances that a single reward model may overlook, we also adopt **GPT-4o** as a Judge (Ouyang et al., 2022). Following the standard protocol (Li et al., 2023), we present GPT-4o with the outputs from T-POP and a baseline, and report the win rate.

During the online interaction phase of our T-POP, we use **GPT-4o** to simulate the user and provide pairwise preference feedback based on the target attribute. The evaluation prompts are adapted from the AlpacaEval standard format.

## 4.2 Main Results

An effective personalization method should generate text that is both **strongly** and **consistently** aligned with user preferences. To ensure a comprehensive evaluation, we assess these two aspects separately. First, we utilize the Reward Model Score (Wang et al., 2024) to quantify the **strength** of personalization (Sec. 4.2.1). Second, to measure **consistency**, we report the win rate against the base LLM in pairwise comparisons judged by GPT-4o (Sec. 4.2.2).

### 4.2.1 ArmoRM Scores: Analysis of the Strength of Personalization

The main quantitative results, presented in Table 1, benchmark T-POP against strong baselines across a wide range of datasets and attributes. The scores in Table 1 underscore the effectiveness of T-POP in achieving stronger alignment A detailed model-by-model analysis reveals that ours algorithm consistently delivers substantial gains over all baselines, including the strongest baseline, AMULET. The performance uplift is most pronounced on Qwen2-7B, where T-POP demonstrates an average improvement of **28.0%** over the second best method, AMULET, across all four preference attributes. This is closely followed by a **19.9%** average gain over AMULET on the Mistral-7B model. On Llama-3.1-8B, the race is highly competitive, with T-POP and AMULET each securing state-of-the-art scores in two of the four preference dimensions; however, T-POP still maintains a marginal edge with a final average score of **0.535** compared to AMULET's **0.5325**. Aggregating these results, T-POP establishes a robust overall average improvement of **14.7%** against AMULET. This persistent and significant performance improvement across diverse models validates the efficacy of our dueling bandit-based test-time personalization framework, which more efficiently captures the nuances of user preferences than other test-time adaptation methods.

Table 1: Score comparison across different datasets, attributes and LLMs. The best score is highlighted in **bold**, and the second best score is highlighted in *italics*.

| Model | Dataset | Creative | | | | | | Verbose | | | | | | Concise | | | | | | Uplifting | | | | | |
|---|---|---|---|---|---|---|---|---|---|---|---|---|---|---|---|---|---|---|---|---|---|---|---|---|---|
| | | Base | Pref | BS16 | LA | Amulet | T-POP | Base | Pref | BS16 | LA | Amulet | Ours | Base | Pref | BS16 | LA | Amulet | T-POP | Base | Pref | BS16 | LA | Amulet | T-POP |
| Mistral-7B | HelpSteer | 0.30 | 0.30 | 0.34 | 0.36 | *0.39* | **0.48** | 0.27 | 0.27 | *0.31* | *0.31* | 0.30 | **0.40** | 0.41 | 0.42 | 0.50 | *0.52* | 0.52 | **0.59** | 0.33 | 0.33 | 0.39 | 0.40 | *0.41* | **0.50** |
| | Personal | 0.34 | 0.34 | 0.35 | 0.38 | *0.42* | **0.47** | 0.30 | 0.30 | 0.30 | 0.30 | 0.30 | **0.39** | 0.47 | 0.49 | 0.50 | *0.54* | 0.53 | **0.65** | 0.41 | 0.42 | 0.42 | 0.45 | *0.46* | **0.52** |
| | Truthful QA | 0.32 | 0.33 | 0.34 | 0.38 | *0.41* | **0.51** | 0.30 | 0.31 | 0.31 | *0.33* | 0.32 | **0.43** | 0.41 | 0.44 | 0.47 | *0.51* | 0.49 | **0.54** | 0.36 | 0.38 | 0.39 | *0.47* | 0.47 | **0.54** |
| | Ultra Chat | 0.34 | 0.35 | 0.35 | 0.36 | *0.38* | **0.47** | 0.31 | 0.31 | 0.31 | 0.32 | 0.31 | **0.39** | 0.45 | 0.46 | 0.47 | 0.49 | *0.51* | **0.61** | 0.38 | 0.39 | 0.39 | 0.41 | *0.42* | **0.50** |
| | Average | 0.32 | 0.33 | 0.34 | 0.37 | *0.40* | **0.48** | 0.30 | 0.30 | 0.31 | *0.32* | 0.31 | **0.40** | 0.43 | 0.45 | 0.48 | *0.52* | 0.51 | **0.60** | 0.37 | 0.38 | 0.40 | 0.43 | *0.44* | **0.51** |
| Qwen2-7B | HelpSteer | 0.34 | 0.34 | 0.35 | 0.35 | *0.36* | **0.50** | 0.31 | 0.32 | *0.33* | *0.33* | 0.30 | **0.44** | 0.43 | 0.48 | 0.50 | 0.57 | *0.59* | **0.60** | 0.38 | 0.38 | 0.39 | 0.39 | *0.41* | **0.52** |
| | Personal | 0.33 | 0.34 | 0.34 | 0.37 | *0.41* | **0.49** | *0.31* | *0.31* | *0.31* | 0.30 | 0.28 | **0.43** | 0.41 | 0.48 | 0.49 | 0.53 | *0.54* | **0.65** | 0.40 | 0.42 | 0.42 | *0.43* | 0.42 | **0.55** |
| | Truthful QA | 0.32 | 0.33 | 0.33 | 0.34 | *0.36* | **0.53** | 0.30 | 0.31 | 0.32 | *0.33* | 0.32 | **0.47** | 0.41 | 0.46 | 0.50 | **0.54** | 0.51 | *0.53* | 0.36 | 0.38 | 0.39 | 0.44 | *0.45* | **0.58** |
| | Ultra Chat | 0.34 | 0.34 | 0.34 | 0.35 | *0.36* | **0.47** | 0.31 | 0.32 | *0.33* | 0.32 | 0.31 | **0.44** | 0.40 | 0.45 | 0.46 | 0.54 | *0.57* | **0.62** | 0.38 | 0.39 | 0.39 | *0.40* | 0.39 | **0.54** |
| | Average | 0.33 | 0.34 | 0.34 | 0.35 | *0.37* | **0.50** | 0.31 | *0.32* | *0.32* | *0.32* | 0.30 | **0.45** | 0.41 | 0.47 | 0.49 | *0.55* | 0.55 | **0.60** | 0.38 | 0.39 | 0.40 | *0.42* | 0.42 | **0.55** |
| Llama-3.1-8B | HelpSteer | 0.33 | 0.34 | 0.36 | 0.44 | *0.50* | **0.51** | 0.30 | 0.31 | 0.33 | 0.36 | *0.41* | **0.51** | 0.40 | 0.43 | 0.45 | 0.53 | *0.57* | **0.62** | 0.36 | 0.37 | 0.39 | 0.45 | *0.50* | **0.53** |
| | Personal | 0.35 | 0.36 | 0.36 | 0.46 | **0.62** | *0.52* | 0.31 | 0.31 | 0.31 | 0.35 | **0.49** | *0.46* | 0.39 | 0.44 | 0.45 | 0.53 | **0.67** | *0.66* | 0.42 | 0.44 | 0.43 | 0.49 | **0.61** | *0.55* |
| | Truthful QA | 0.31 | 0.33 | 0.33 | 0.41 | **0.56** | *0.52* | 0.29 | 0.29 | 0.31 | 0.34 | *0.44* | **0.54** | 0.37 | 0.40 | 0.42 | 0.49 | **0.52** | *0.51* | 0.34 | 0.36 | 0.37 | 0.43 | *0.49* | **0.53** |
| | Ultra Chat | 0.33 | 0.34 | 0.34 | 0.42 | **0.57** | *0.50* | 0.31 | 0.32 | 0.32 | 0.36 | *0.41* | **0.49** | 0.38 | 0.41 | 0.41 | 0.48 | *0.53* | **0.60** | 0.37 | 0.38 | 0.38 | 0.44 | *0.48* | **0.52** |
| | Average | 0.33 | 0.34 | 0.35 | 0.43 | **0.58** | *0.51* | 0.30 | 0.31 | 0.32 | 0.35 | *0.44* | **0.50** | 0.38 | 0.42 | 0.43 | 0.51 | *0.57* | **0.60** | 0.37 | 0.39 | 0.39 | 0.45 | **0.54** | *0.53* |

Furthermore, we analyze the impact of the number of user interactions (iterations) on the performance of T-POP. To demonstrate the robustness of its learning efficiency, we present results from two distinct experimental settings: the concise attribute on the Personal dataset and the HelpSteer dataset (Fig. 2). As illustrated across both figures, all three models—Llama-3.1-8B, Mistral-7B, and Qwen2-7B—exhibit a remarkably consistent and efficient learning curve. **The reward scores increase sharply within the first 20 iterations in both scenarios**, indicating that T-POP rapidly captures user preferences with minimal feedback, regardless of the specific task. Following this initial surge, performance gains begin to plateau, with the models reaching their peak alignment between 40 and 60 interactions. Subsequently, the scores remain stable or decrease slightly, which can be attributed to potential overfitting. This consistent trend of rapid initial improvement followed by convergence across diverse datasets further validates the data efficiency and swift personalization capability of T-POP.

### 4.2.2 WIN RATE: ANALYSIS OF THE CONSISTENCY OF PERSONALIZATION

To assess the **consistency** of our personalization method, we employ GPT-4o as a judge to perform pairwise comparisons. For each prompt, GPT-4o evaluates which of two responses—one from our method and one from the base LLM—is better aligned with a given personalization attribute. Table 2 presents the results, where each value represents the *win rate* against the base LLM. This metric measures how consistently an algorithm produces a qualitatively superior and personalized response.

The results show that T-POP achieves personalization with remarkable consistency. Across the 36 experimental settings (3 LLMs × 4 attributes × 3 datasets), our T-POP achieves the highest or second-highest average win rate in 31 cases. Crucially, the win rate for T-POP is almost universally above 90%, averaging **94.2%** across all settings. A win rate over 90% signifies a high degree of confidence that T-POP consistently provides correct alignment and personalization, leading to responses that are qualitatively superior to those from the unguided base model. This robust performance indicates that our T-POP is not only powerful but also highly reliable.

In summary, the ArmoRM scores in Table 1 and the win rates in Table 2 jointly demonstrate that **T-POP achieves strong and consistent personalization**.

## 5 ABLATION STUDY

**The Impact of Reward Weight** $w$**.** Fig. 3 illustrates the performance of T-POP across a range of $w$ values for all three backbone models. The results exhibit a clear and consistent trend. At $w = 0.0$, where T-POP effectively deactivates the personalization component, the reward scores are at their lowest, representing the performance of the base LLM. A sharp and substantial improvement in the

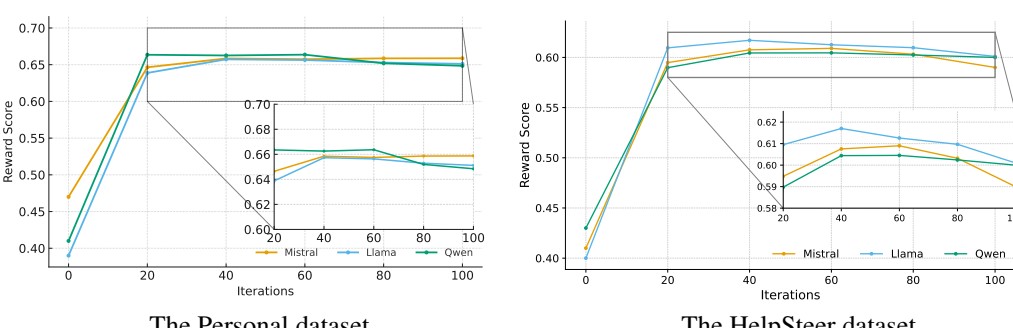

Figure 2: The effect of the number of user interactions on the Reward Score for different models. The results correspond to the concise attribute.

Table 2: Win rate of different algorithms against the base LLM in terms of personalization. The best score is highlighted in **bold**, and the second best score is highlighted in *italics*.

| Model | Dataset | Creative | | | | | Verbose | | | | | Concise | | | | | Uplifting | | | | |
|---|---|---|---|---|---|---|---|---|---|---|---|---|---|---|---|---|---|---|---|---|---|
| | | Pref | BS16 | LA | Amulet | T-POP | Pref | BS16 | LA | Amulet | Ours | Pref | BS16 | LA | Amulet | T-POP | Pref | BS16 | LA | Amulet | T-POP |
| Mistral-7B | HelpSteer | 95.5% | 94.0% | *98.1%* | 90.2% | **99.5%** | 79.4% | 76.5% | *91.0%* | 79.7% | **93.1%** | 87.9% | *89.5%* | 87.9% | 75.1% | **92.4%** | 86.7% | 85.2% | *95.3%* | 92.3% | **97.2%** |
| | Personal | 97.1% | 94.3% | *98.5%* | 96.6% | **99.1%** | 85.4% | 75.4% | **96.5%** | 87.8% | *92.5%* | 95.4% | 94.0% | 93.8% | 71.3% | **96.7%** | 86.4% | 85.6% | *94.2%* | 90.2% | **98.2%** |
| | Truthful QA | 85.4% | 83.0% | *94.5%* | 93.3% | **99.6%** | 79.1% | 77.5% | *90.1%* | 78.7% | **95.4%** | 77.4% | **80.5%** | 70.3% | 70.8% | *72.3%* | 85.8% | 82.9% | *91.9%* | 88.2% | **96.5%** |
| | Average | 92.6% | 90.4% | *97.0%* | 93.4% | **99.4%** | 81.3% | 76.5% | *92.5%* | 82.1% | **93.7%** | 86.9% | **88.0%** | 84.0% | 72.4% | *87.7%* | 86.3% | 84.6% | *93.8%* | 90.2% | **97.3%** |
| Qwen2-7B | HelpSteer | *94.0%* | 92.8% | 86.1% | 89.9% | **96.6%** | *93.2%* | 90.9% | 82.9% | 83.3% | **94.0%** | 88.2% | 89.5% | **92.8%** | 91.6% | *92.0%* | 83.8% | 83.5% | 75.2% | **97.7%** | *98.2%* |
| | Personal | 95.2% | 96.3% | *98.2%* | 96.8% | **99.1%** | 93.3% | *96.7%* | **100%** | 73.7% | 90.3% | 95.8% | 97.6% | **99.1%** | 92.0% | *98.7%* | 83.5% | *89.9%* | **95.8%** | 91.1% | 91.3% |
| | Truthful QA | *90.1%* | 85.4% | 88.2% | 83.9% | **98.5%** | 78.2% | 79.5% | *84.3%* | 78.9% | **96.0%** | 88.9% | 90.1% | **92.2%** | *91.2%* | 79.0% | 81.9% | 81.2% | 80.8% | *93.0%* | **99.1%** |
| | Average | *93.1%* | 91.5% | 90.8% | 90.2% | **98.1%** | 88.2% | 89.0% | *89.1%* | 78.6% | **93.4%** | 91.0% | 92.4% | **94.7%** | *91.6%* | 89.9% | 83.1% | 84.9% | 83.9% | *93.9%* | **96.2%** |
| Llama-3.1-8B | HelpSteer | 97.4% | 96.2% | 97.4% | *97.6%* | **98.6%** | 91.7% | 91.4% | **97.6%** | 94.7% | *97.6%* | 89.0% | 89.3% | **94.3%** | 86.3% | *92.3%* | 89.4% | 88.8% | **99.0%** | 97.5% | *97.6%* |
| | Personal | 96.3% | 95.1% | 97.1% | **99.8%** | *98.9%* | 91.4% | 90.6% | 93.8% | **99.6%** | *94.5%* | 96.2% | 97.0% | 97.2% | *97.3%* | **97.4%** | 94.1% | 94.0% | *99.6%* | **100%** | 94.0% |
| | Truthful QA | 94.1% | 92.3% | 97.2% | **99.5%** | *97.3%* | 87.3% | 86.7% | **96.5%** | 93.2% | *95.4%* | 71.9% | 76.9% | 74.7% | **85.5%** | 68.8% | 82.7% | 82.6% | **95.3%** | 92.8% | *93.5%* |
| | Average | 95.9% | 94.5% | 97.2% | **99.0%** | *98.3%* | 90.1% | 86.2% | **96.0%** | 95.8% | *95.8%* | 85.7% | 87.7% | 87.7% | **89.7%** | 86.1% | 88.7% | 88.5% | **97.8%** | 96.8% | 95.0% |

reward scores is observed across all models at $w = 0.1$, and the performance peaks at $w = 1.0$. This indicates that a moderate reward signal is highly effective at steering the generation towards the user preference. However, as the weight is further increased to $w = 2.0$ and subsequently to $w = 5.0$, the reward scores show a noticeable decline. This suggests that an excessively high reward weight can be counterproductive. This is likely because an overly strong preference signal begins to *interfere with the inherent generation capabilities of the backbone model*, $\pi_{base}$, leading the decoding strategy to myopically optimize for the reward. This can result in outputs that, while superficially aligned, may lack coherence or quality. This phenomenon is often referred to as reward hacking. Our findings suggest that an optimal value for $w$ lies in the vicinity of $w = 1.0$, which strikes an effective balance between personalization strength and the preservation of generation quality.

**Impact of Model Size.** To assess the scalability and model-agnostic properties of T-POP, we evaluate its performance on smaller, resource-efficient LLMs. Specifically, we apply T-POP to Qwen2-0.5B-Instruct and Llama-3.2-1B-Instruct, comparing the ArmoRM scores against that of the base models. The results are presented in Table 3, which confirm that T-POP is able to effectively personalize these smaller models. Notably, T-POP delivers a substantial improvement for the Llama-3.2-1B-Instruct model, increasing its alignment score from 0.28 to 0.44. This finding has significant implications, as it demonstrates that our method can dramatically enhance the capabilities of smaller models, enabling them to achieve a level of personalization typically associated with much larger models. This highlights the potential of T-POP for applications with constrained computational resources, such as on-device deployment.

**Impact of the Uncertainty Bonus.** We perform the experiments using the Llama-3.1-8B-Instruct backbone on the Personal (Gao et al., 2024) dataset for the "concise" attribute, and the model is

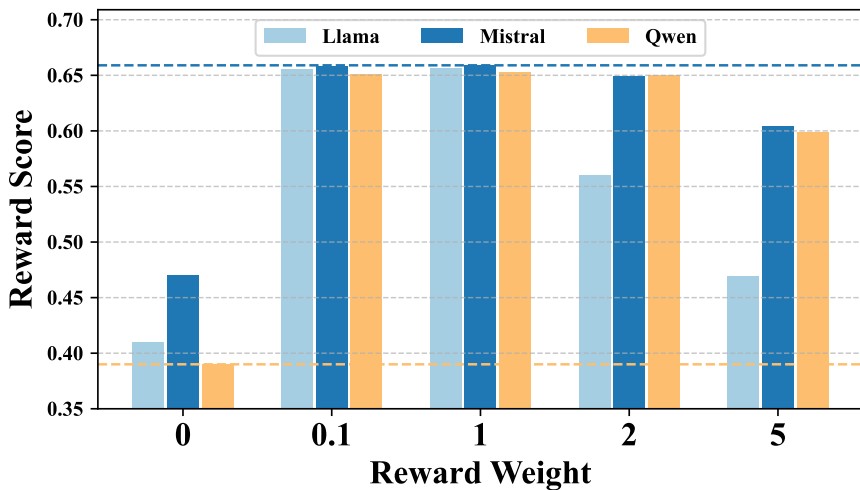

Figure 3: The effect of the reward weight ($w$) on the alignment performance of `T-POP` across three different backbone models.

Table 3: ArmoRM scores of our `T-POP` for models with different sizes.

| Model | Base Score | `T-POP` Score |
|---|---|---|
| Qwen2-0.5B-Instruct | 0.27 | **0.29** |
| Qwen2-7B-Instruct | 0.37 | **0.51** |
| Llama-3.2-1B-Instruct | 0.28 | **0.44** |
| Llama-3.1-8B-Instruct | 0.35 | **0.55** |

trained online for 20 iterations. We compare `T-POP` against three alternative strategies for the exploration arm, replacing our metric with different heuristics:

- **Entropy Bonus:** We replaced our uncertainty metric (equation 5) with a token-level entropy term, $\text{Bonus}(v) = -P(v)\log P(v)$, where $P(v)$ is the softmax probability of the token scores.
- **Boltzmann Exploration:** A standard reinforcement learning baseline representing "noisy exploitation." The exploration arm employs high-temperature sampling ($T = 1.5$) on the reward-guided logits: $v_{p,2} \sim \text{Softmax}\left(\frac{\log \pi_{base} + \omega \cdot r}{T}\right)$.
- **Random:** The exploration arm is generated via random sampling from the base LLM $\pi_{base}$, serving as a performance lower bound.

Table 4: Ablation study on exploration strategies.

| Method | Final Score | Improvement vs. Random |
|---|---|---|
| T-POP (Random) | 0.51 | - |
| T-POP (Entropy) | 0.53 | +0.02 |
| T-POP (Boltzmann) | 0.57 | +0.06 |
| **T-POP (Ours)** | **0.64** | **+0.13** |

The results are shown in Table 4, which confirm the validity of our algorithm: strategies like Entropy and Boltzmann Exploration primarily leverage the *aleatoric uncertainty* (ambiguity inherent in the next-token prediction of the language model). In contrast, the uncertainty metric employed by `T-POP` utilizes the gradient norm to capture the *epistemic uncertainty* regarding the user's preference parameters (Verma et al., 2024). To efficiently solve the cold-start problem, the system must explore regions where the *reward model* lacks knowledge, not merely where the *language model* is diverse. This theoretical distinction translates directly into the superior data efficiency observed in our method.

**Alignment-Compute Trade-off.** To rigorously evaluate the computational cost, we measured the wall-clock inference time for `T-POP` against the state-of-the-art baseline, AMULET.

Table 5: Wall-clock inference time comparison (seconds).

| Method | AMULET | T-POP (Ours) |
|---|---|---|
| Query-level Latency | 11.25 | 23.26 |
| Token-level Latency | 0.09 | 0.18 |

As presented in Table 5, `T-POP` incurs approximately twice the latency of AMULET but remains within the same order of magnitude. This reflects the inherent *alignment-compute trade-off* noted in prior work (Khanov et al., 2024). We argue this moderate computational cost is justified by the substantial performance gains, as `T-POP` establishes a robust overall average improvement of 14.7% over the strongest baselines in Table1.

## 6 RELATED WORK

### 6.1 ALIGNMENT THROUGH REINFORCEMENT LEARNING FROM HUMAN FEEDBACK

Reinforcement Learning from Human Feedback (RLHF) (Christiano et al., 2017; Ziegler et al., 2019) is the standard paradigm for aligning LLMs with human preferences. The canonical pipeline (Ouyang et al., 2022) involves three stages: 1) supervised fine-tuning (SFT) on high-quality demonstrations; 2) training a reward model (RM) (Stiennon et al., 2020) on a dataset of human-ranked responses; and 3) fine-tuning the SFT model using an RL algorithm such as PPO (Schulman et al., 2017), with the RM providing the reward signal. The computational expense and instability of PPO-based RLHF have motivated simpler alternatives. For example, direct Preference Optimization (DPO) (Rafailov et al., 2023) bypasses explicit reward modeling by reframing alignment as a direct policy optimization problem. However, these advancements still produce a single, static policy aligned with a pre-collected, offline dataset, often scaled with techniques like RLAIF (Bai et al., 2022).

### 6.2 PERSONALIZED ALIGNMENT

Since the universal preference model of conventional RLHF is ill-suited for personalization, a dedicated research area has emerged to adapt LLMs to individual users. One approach involves creating large-scale datasets to model diverse preferences by mapping sociodemographics (PRISM (Kirk et al., 2024)) or constructing user personas from psychological traits (ALIGNX (Li et al., 2025), PAPI (Zhu et al., 2025)). A more data-efficient direction models preferences in a compact, low-dimensional latent space, for instance, by representing them as a linear combination of base reward functions (PReF (Shenfeld et al., 2025), multi-objective alignment (Zhou et al., 2023)) or as latent distributions for few-shot adaptation (VPL (Poddar et al., 2024)). The third direction, most aligned with our work, focuses on lightweight, inference-time adaptation of frozen LLMs. These methods steer the decoding process by manipulating the LLM outputs (PAD (Chen et al., 2024), LA (Gao et al., 2024), decoding-time realignment (Liu et al., 2024)), reframing token generation as an online learning problem (AMULET (Zhang et al., 2025b)), or directly modifying the internal states of the LLMs such as the attention head activations (PAS (Zhu et al., 2025)).

## 7 CONCLUSION

In this work, we addressed the critical cold-start problem in personalizing LLMs for new users. We introduced `T-POP`, a novel algorithm that enables rapid, real-time personalization by learning directly from online pairwise preference feedback. By synergistically integrating test-time alignment with dueling bandits, `T-POP` steers the decoding process of a frozen LLM to simultaneously exploit learned preferences and efficiently explore for new ones. Our extensive experiments demonstrate that `T-POP` achieves significant performance gains over existing baselines with minimal user interaction, confirming its data efficiency and effectiveness for swift personalization. Future work could explore extending this framework to handle more complex feedback structures or adapt to long-term shifts in user preferences.

## REPRODUCIBILITY STATEMENT

To ensure reproducibility, we have clearly described the detailed experimental setting in Sec. 4.1 and App. B. We have also included important prompts adopted by our algorithm in App. B.

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

## A    STATEMENT ON LLM USAGE

The authors utilized LLMs solely as writing assistants to improve the grammar, clarity, and readability of this paper. All intellectual contributions, including the core ideas, methodology, and analysis of results, were conducted by the human authors.

## B    MORE DETAILS ON THE EXPERIMENTAL SETTING

**Reward Model Architecture.**    The lightweight reward model, $r(\cdot; \theta)$, is implemented as a simple Multi-Layer Perceptron (MLP) head. This network takes the final hidden-state embeddings from the backbone LLM for a given sequence as input. The MLP consists of one hidden layer with a size of 1024, and all hidden layers utilize the ReLU activation function.

**Diagonal Approximation.**    Following the common practice in neural bandits, we use diagonal approximation to approximate hte (Zhang et al., 2021; Zhou et al., 2020)

**Datasets Description.**    Since `T-POP` is a training-free framework, we use the collected data solely for evaluation purposes. Our evaluation suite is constructed from four established benchmarks, from which we only use the question (and discard the responses) to simulate real-world user interactions. The datasets and their sizes are as follows:

- **HelpSteer** (Wang et al., 2023) is a QA dataset aimed at evaluating the model's capability to follow multi-faceted instructions; we utilize its 1,236 testing instances (Zhang et al., 2025b).
- **UltraFeedback** (Cui et al., 2024) is a comprehensive, high-quality AI feedback dataset. From this, we selected two subsets: **Truthful QA** (Lin et al., 2021), using its 811 testing problems to assess factuality, and **UltraChat**, from which we extracted 3,845 problems to evaluate conversational ability (Zhang et al., 2025b).
- **Personal Preference Eval** (Personal) (Gao et al., 2024) is used to evaluate user preference alignment; we utilized the original dataset containing 548 testing instances (Zhang et al., 2025b).

**Hyperparameters.**    The key hyperparameters used for the training of the reward model and the dueling bandit component of `T-POP` throughout our experiments are listed in Table 6.

Table 6: Hyperparameter settings for `T-POP`.

| Category | Hyperparameter | Value |
|---|---|---|
| *Dueling Bandit Parameters* | Reward weight ($w$) | 1.0 |
| | Exploration parameter ($\nu$) | 0.5 |
| | Regularization parameter ($\lambda$) | 1.0 |
| *Reward Model Online Training* | Optimizer | AdamW |
| | Batch size | 8 |
| | Learning rate | 5e-4 |
| | Epochs per query | 100 |
| | Training Iteration | 100 |
| | Weight decay schedule | $1/(N+50)$ |
| *Decoding Parameters* | Max new tokens | 128 |
| | Candidate tokens ($k$) | 40 |

For the weight decay schedule, $N$ denotes the number of training data points.

**Embedding and Judge Models.**    The reward model utilizes embeddings from the `Qwen/Qwen3-Embedding-0.6B` model (Zhang et al., 2025a). For all experiments requiring preference evaluation, including the simulation of user feedback during the online learning phase and the final win-rate judgments, we employ `openai/GPT-4o` (OpenAI et al., 2024).

**Judgement Prompt Template.** To ensure a consistent and reproducible method for both simulating user feedback and performing the final evaluation, we utilized a structured prompt template adapted from the AlpacaEval format (Li et al., 2023). Fig. 4 shows the natural language descriptions for the four core preference attributes used in our experiments. These descriptions serve as the concrete personalization goal.

Figu. 5 displays the main judgment prompt template provided to GPT-4o. In practice, a specific attribute description from Fig. 4 is inserted into the `{attribute description}` field of the main template. The complete prompt then instructs GPT-4o to act as an AI assistant and select which of the two provided responses better embodies the target attribute. This mechanism was used for two critical functions: (1) to generate the online pairwise preference feedback required by `T-POP` during its learning phase, and (2) to conduct the final win-rate evaluations against baseline models, as presented in Section 4.2.

---

**Preference Attribute Descriptions**

PREFERENCE ATTRIBUTES :
"creative": "Your answer should be creative as much as possible.",
"verbose": "Your answer should be verbose as much as possible.",
"concise": "Your answer should be concise as much as possible.",
"uplifting": "Your answer should be uplifting as much as possible."

---

Figure 4: Natural language descriptions for the personalized preference attributes.

---

**GPT-4o Judgement Prompt Template**

Input:
You are an AI assistant that helps determine which response better aligns with a given attribute preference. Given a specific attribute preference, select the response from assistant A or B that better embodies this attribute.Focus on how well each response aligns with the specified attribute, not general quality. Declare your choice by using the format: "[[A]]" if you believe assistant A's response better aligns with the attribute, or "[[B]]" if assistant B's response better aligns with the attribute.
[Target Attribute]
{attribute}: {attribute description}
[User Question]
{query}
[The Start of Assistant A's Answer]
{response 1}
[The End of Assistant A's Answer]
[The Start of Assistant B's Answer]
{response 2}
[The End of Assistant B's Answer]
[Task] Which response better aligns with the "{attribute}" attribute? Consider how well each response embodies the characteristic described above.

Output:
[[A]] or [[B]]

---

Figure 5: The prompt template used to instruct GPT-4o for preference simulation and win rate evaluation.

# C MORE ABLATION EXPERIMENT RESULTS

In this section, we provide additional ablation studies to further validate the efficiency and effectiveness of T-POP. Unless otherwise stated, all experiments in this section are conducted using the **Llama-3.1-8B-Instruct** backbone on the **Personal** dataset for the **concise** attribute.

## C.1 ANALYSIS OF COLD-START PERFORMANCE (EARLY ITERATIONS)

To rigorously evaluate T-POP's capability in addressing the cold-start problem, we analyzed its performance at extremely early stages of user interaction ($T = 5$ and $T = 10$). Table 7 compares the ArmoRM scores of T-POP against baselines.

Remarkably, with only **5 user interactions**, T-POP achieves a reward score of 0.56, which already surpasses the strong training-free baseline Linear Alignment (LA, 0.53) and significantly outperforms static methods like Prompting (0.44). By $T = 10$, the performance gap further widens, demonstrating T-POP's ability to rapidly adapt to user preferences with minimal data.

Table 7: Performance comparison at early interaction stages (Proof of Rapid Adaptation).

| Method | Base | Pref | BS16 | LA | Amulet | T-POP (Iter=5) | T-POP (Iter=10) | T-POP (Converged) |
|---|---|---|---|---|---|---|---|---|
| **ArmoRM Score** | 0.39 | 0.44 | 0.45 | 0.53 | **0.67** | 0.56 | 0.63 | *0.66* |

## C.2 ABLATION ON EXPLORATION STRATEGIES

A key component of T-POP is the construction of the "Exploration Sequence" ($y_{t,2}$) using a principled uncertainty bonus. To justify our design choice, we compared T-POP against three variant exploration strategies:

- **Variant A: Entropy Bonus.** We replaced our uncertainty metric with a token-level entropy term: $\text{Bonus}(v) = -P(v) \log P(v)$, targeting tokens with high predictive uncertainty in the base model.
- **Variant B: Boltzmann Exploration.** Instead of an explicit bonus, we employed High-Temperature Sampling ($T_{high} = 1.5$) on the reward-guided logits to induce "noisy exploitation." The token selection follows:

$$v_{p,2} \sim \text{Softmax} \left( \frac{\log \pi_{base}(\cdot|y_{t,2}) + \omega \cdot r([y_{t,2}, \cdot]; \theta_t)}{T_{high}} \right)$$

- **Variant C: Random.** The exploration arm is generated via random sampling from the base LLM, serving as a lower bound.

As shown in Table 8, our uncertainty-based approach significantly outperforms heuristic methods (Entropy) and noisy sampling (Boltzmann). This confirms that estimating epistemic uncertainty via the Fisher Information Matrix provides a more informative signal for the reward model than simple aleatoric uncertainty or randomness.

Table 8: Ablation study on different exploration strategies (Iteration 20).

| Method | Final Score (Iter=20) | Improvement over Random |
|---|---|---|
| T-POP (Random) | 0.51 | - |
| T-POP (Entropy) | 0.53 | +0.02 |
| T-POP (Boltzmann) | 0.57 | +0.06 |
| **T-POP (Ours)** | **0.64** | **+0.13** |

## C.3 INFERENCE LATENCY ANALYSIS

We further evaluated the computational overhead of T-POP compared to the SOTA baseline AMULET. Table 9 reports the wall-clock inference time per query and per token.

**T-POP vs. AMULET:** As shown in Table 9, T-POP incurs a higher latency compared to AMULET. This overhead primarily stems from the embedding phase and the forward pass of the lightweight Reward Model (RM) during the decoding process. However, importantly, our token-level latency (0.18s) remains within the same order of magnitude as AMULET (0.09s), making it practically feasible for real-time user interactions.

**Trade-off Justification:** As noted in prior work on test-time alignment (Khanov et al., 2024), there is an inherent "Computation vs. Alignment" trade-off. Given that T-POP effectively addresses the Cold-Start Problem, enabling personalization for new users where static baselines fail, we argue that this marginal increase in computational cost is a justified investment for the significant gains in alignment quality and data efficiency.

Table 9: Wall-Clock Inference Time Comparison.

| Metric | AMULET | T-POP (Ours) |
|---|---|---|
| Query-level Latency | 11.25 s | 23.26 s |
| Token-level Latency | 0.09 s | 0.18 s |

