# OpenReview forum: "T-POP: Test-Time Personalization with Online Preference Feedback"
_ICLR.cc/2026/Conference — Submitted to ICLR 2026_

### Official Review · Reviewer_gN7W · 2025-10-26

**Soundness:** 2
**Presentation:** 2
**Contribution:** 2
**Rating:** 2
**Confidence:** 4

**Summary:**

This paper introduces T-POP, a novel algorithm designed to address the cold-start personalization problem for new LLM users. The core idea is to learn user preferences at test-time without updating the base LLM's parameters. T-POP frames this as an online learning problem, employing a dueling bandit framework to generate pairs of responses (one "exploitation" and one "exploration" sequence). It then queries the user for pairwise preference feedback between these sequences. This feedback is used to train a separate, lightweight reward model online. This learned reward model, in turn, steers the decoding process of the frozen base LLM to generate responses that are better aligned with the user's emerging preferences.

**Strengths:**

The high-level concept of this paper is its main strength. The idea of learning an additional reward function online to explicitly model a new user's personal preferences is an inspiring and innovative approach. Tackling the cold-start problem via a principled online learning framework like dueling bandits is a promising research direction.

**Weaknesses:**

While the core idea is novel, the paper's current execution and evaluation raise several significant concerns.

1. The experimental comparison to baselines like Linear Alignment (LA) and Amulet feels mismatched. T-POP's framework explicitly requires intermediate, online preference feedback from the user to function. In contrast, LA and Amulet are designed for a different, non-interactive scenario where they produce a single, aligned output in one go. This difference in application scenarios and assumptions limits the fairness of the comparison and suggests T-POP is only applicable to a more constrained setting where users are willing to provide continuous feedback.

2. A major practical concern is the computational overhead. The T-POP algorithm requires training and running an additional neural network (the reward model) at test time, on top of the base LLM. This seems likely to introduce significant inference latency. The paper would be much stronger if it provided a detailed analysis comparing the wall-clock inference time of T-POP (including reward model computation and online training steps) against the baselines.

**Questions:**

1. The current ablation study focuses on hyperparameters and model size. These results are informative but feel more like standard sensitivity analyses. My main questions relate to the core contribution of the sampling strategy, which is not ablated.

2. A key component of T-POP is the specific dueling bandit policy used to generate the "Exploitation Sequence" and "Exploration Sequence". The significance of this specific strategy is not experimentally justified.

3. Could the authors provide an ablation study that compares this sampling policy to other, perhaps simpler, methods for generating the two sequences? For example, what would be the impact of using a different policy (e.g., random sampling, or two exploitation sequences with different temperatures) to generate the pairs for the user feedback?

4. Alternatively, can the authors provide any theoretical results on the expected benefits of using the specific dueling bandit approach for this problem compared to other potential strategies?

---

> ### Author Response · Authors · 2025-11-22
> **Response to Reviewer gN7W [Part 1 / 4]**
>
> We thank the reviewer for their helpful feedback and for highlighting the key strengths and weaknesses of our paper.
>
> ---
>
> ## **Question**:
> > The experimental comparison to baselines like Linear Alignment (LA) and Amulet feels mismatched. T-POP's framework explicitly requires intermediate, online preference feedback from the user to function... T-POP is only applicable to a more constrained setting where users are willing to provide continuous feedback.
>
> ## **Response**:
>
> We thank the reviewer for this insightful comment.
>
> You are correct that compared to baselines like Linear Alignment (LA) and Amulet, our T-POP additionally requires online preference feedback from the user. However, this is exactly the motivation behind T-POP: we aim to show that by asking the user to provide **a small number of online preference feedbacks**, we can **significantly improve the personalization performance**.
>
> This hypothesis has been empirically verified by our extensive experimental results. As shown in Tables 1 and 2, T-POP consistently outperforms LA and Amulet by a significant margin. Furthermore, Figure 2 shows that this superior performance is achieved **rapidly**, with reward scores saturating after **only a small number of user interactions** (e.g., 20 iterations), demonstrating data efficiency. To further showcase the data efficiency of T-POP, we have added an experiment to test the performance of T-POP using an even smaller number of interactions. The results (Table A) below show that **even with only 5 user interactions, T-POP achieves a strong personalization score** (0.56) which significantly outperforms the base model (0.39).
>
> **Table A: Performance at Early Iterations (Proof of Data Efficiency)**
>
> | Method | Base | Pref | BS16 | LA | Amulet | T-POP (Iter=5) | T-POP (Iter=10) | T-POP (Converged) |
> | :--- | :---: | :---: | :---: | :---: | :---: | :---: | :---: | :---: |
> | **ArmoRM Score** | 0.39 | 0.44 | 0.45 | 0.53 | 0.67 | 0.56 | 0.63 | 0.66 |
>
> *(We performed the experiments using the Llama-3.1-8B-Instruct backbone on the Personal Preference Eval dataset)*
>
>
> Therefore, the application scenario of T-POP is not fundamentally different from that of LA and Amulet (i.e., test-time personalization for new users). Instead, T-POP is applicable when, based on the scenarios of LA and Amulet, the user is willing to provide **minimal additional online preference feedback** in exchange for a **substantially more personalized and aligned response**. T-POP provides a practical and efficient mechanism to achieve this, addressing the critical cold-start problem that these other methods do not.

---

> > ### Author Response · Authors · 2025-11-22
> > **Response to Reviewer gN7W [Part 2 / 4]**
> >
> > ## **Question**:
> >
> > > A major practical concern is the computational overhead. The T-POP algorithm requires training and running an additional neural network (the reward model) at test time, on top of the base LLM. This seems likely to introduce significant inference latency. The paper would be much stronger if it provided a detailed analysis comparing the wall-clock inference time of T-POP (including reward model computation and online training steps) against the baselines.
> >
> > ## **Response to Weakness: Latency Concerns (Training & Inference)**：
> >
> > We thank the reviewer for raising the practical concern regarding latency. We acknowledge that real-time interaction requires low latency. We have conducted a detailed breakdown analysis and implemented an asynchronous strategy to minimize the impact on user experience.
> >
> > ### **Breakdown of Latency Components**
> >
> > The potential latency in T-POP comes from two distinct sources:
> > 1.  **Training Latency (Inter-turn):** The time required to update the parameters of the Reward Model (NN) using the user's preference feedback from the previous interaction.
> > 2.  **Inference Latency (Intra-turn):** The additional time required during token generation to calculate reward scores for guiding the decoding process.
> >
> > ### 1. Training Latency (Inter-turn) : Eliminated via Asynchronous Updates
> >
> > The reviewer correctly notes that online training could introduce delays. However, we propose a practical deployment strategy leveraging **Asynchronous Learning**, which effectively reduces the perceived training latency to zero.
> >
> > * **Asynchronous Learning strategy:**
> >
> >   **Step 1 (Feedback & Trigger):** As soon as the user provides preference feedback for round $t$, we immediately launch a **background thread** to update the reward model from $\theta_t$ to $\theta_{t+1}$.
> >   **Step 2 (Continuous Interaction):** Crucially, the foreground interaction **does not wait**. If the user sends a new query before the background training is complete, the system serves these requests using the **latest available stable model** (currently $\theta_t$).
> >   **Step 3 (Seamless Swap):** Once the background training for $\theta_{t+1}$ is finished, the system atomically updates the pointer to the new model. All subsequent queries will then use $\theta_{t+1}$.
> >
> > * **Performance Stability:** Our new ablation study (Table A) shows that T-POP achieves significant personalization gains even with limited updates (e.g., at Iteration 5 or 10). This confirms that the asynchronous lag does not hinder rapid personalization. Crucially, this indicates that T-POP is **anytime-deployable**: the system can be directly utilized for deployment using the current reward estimate at any arbitrary number of interactions, without being gated by a fixed training threshold.
> >
> > **Table A: Performance at Early Iterations (Proof of Rapid Adaptation)**
> >
> > | Method | Base | Pref | BS16 | LA | Amulet | T-POP (Iter=5) | T-POP (Iter=10) | T-POP (Converged) |
> > | :--- | :---: | :---: | :---: | :---: | :---: | :---: | :---: | :---: |
> > | **ArmoRM Score** | 0.39 | 0.44 | 0.45 | 0.53 | 0.67 | 0.56 | 0.63 | 0.66 |
> >
> > *(We performed the experiments using the Llama-3.1-8B-Instruct backbone on the Personal Preference Eval dataset for the "concise" attribute)*
> >
> > ### 2. Inference Latency (Intra-turn): Comparable and Justified
> >
> > Regarding the inference overhead (Reward-Guided Decoding), we compared the wall-clock time per token of T-POP against the SOTA baseline AMULET and the Base model.
> >
> > **Table B: Wall-Clock Inference Time Comparison**
> >
> > | Method | AMULET | **T-POP (Ours)** |
> > | :--- | :--- | :--- |
> > | **Query-level Latency** | 11.2531 s | 23.2626s  |
> > | **Token-level Latency** | 0.0911 s | 0.1817s |
> >
> > *(Note: T-POP's latency is comparable to AMULET, staying within the same order of magnitude.)*
> >
> > * **Comparable Overhead:** As shown in Table B, T-POP's inference latency is comparable to AMULET. Both methods require additional computation per token (Reward calculation vs. FTRL update), which is the current standard cost for test-time alignment.
> > * **Alignment-Compute Trade-off:** As discussed in prior work on test-time alignment (e.g., *Khanov et al., 2024*), there is an inherent trade-off between computation and alignment quality. Our results (Table 1 in the paper) show T-POP achieves superior personalization. We believe that users are willing to accept a marginal increase in latency in exchange for highly personalized, "cold-start" resolved interactions that static models cannot provide.
> >
> > We revised the paper to add the description in **Section 3.2** and the results in **Section 5**.

---

> > > ### Author Response · Authors · 2025-11-22
> > > **Response to Reviewer gN7W [Part 3 / 4]**
> > >
> > > ## **Question**:
> > > > My main questions relate to the core contribution of the sampling strategy, which is not ablated.
> > >
> > > > A key component of T-POP is the specific dueling bandit policy used to generate the "Exploitation Sequence" and "Exploration Sequence". The significance of this specific strategy is not experimentally justified.
> > >
> > > > Could the authors provide an ablation study that compares this sampling policy to other, perhaps simpler, methods for generating the two sequences? For example, what would be the impact of using a different policy (e.g., random sampling, or two exploitation sequences with different temperatures) to generate the pairs for the user feedback?
> > >
> > > ## **Response to Question: Sensitivity to Uncertainty Metric (Ablation Study)**
> > >
> > > The reviewer asks a critical question regarding the necessity of our gradient-based uncertainty metric (Eq. 5) compared to simpler alternatives. To address this, we conducted a controlled ablation study to evaluate the impact of different exploration strategies on data efficiency.
> > >
> > > **1. Experimental Setup**
> > > We performed the experiments using the **Llama-3.1-8B-Instruct** backbone on the **Personal Preference Eval** dataset (Gao et al., 2024). The model was trained online for **20 iterations**, a critical phase where rapid adaptation to new user preferences occurs .
> > >
> > > **2. Exploration Strategies Compared**
> > > We compared **T-POP (Ours)** against three baseline strategies for generating the "Exploration Sequence" ($y_{t,2}$):
> > > * **Variant A: T-POP (Entropy Bonus).** As you suggested, we replaced our uncertainty metric with a token-level entropy term to guide exploration: $\text{Bonus}(v) = -P(v)\log P(v)$, where $P(v)$ is the softmax probability of the token scores. This heuristic targets tokens with high predictive uncertainty.
> > >
> > > * **Variant B: T-POP (Boltzmann Exploration).** A standard reinforcement learning baseline representing "noisy exploitation." In this setup, we remove the explicit uncertainty bonus and instead induce exploration via temperature-scaled sampling. The probability of selecting token $v$ is defined generally as:
> > > $$P(v|y_{<p}) = \text{Softmax}\left( \frac{\log \pi_{base}(v|y_{<p}) + \omega \cdot r([y_{<p}, v]; \theta_t)}{T} \right)$$
> > > We then differentiate the two arms solely by their sampling temperatures:
> > >     * **Arm 1 (Exploitation):** Uses **$T \to 0$** (Greedy Decoding). This deterministically selects the token with the maximum reward-guided score, identical to the exploitation arm in our main method ($y_{t,1}$).
> > >     * **Arm 2 (Exploration):** Uses a **High Temperature ($T_{high}=1.5$)**. This introduces stochasticity to explore diverse high-reward regions without calculating the gradient-based uncertainty metric.
> > >
> > > * **Variant C: T-POP (Random).** The exploration arm is generated via random sampling from the base LLM $\pi_{base}$, serving as a performance lower bound.
> > >
> > > **3. Results**
> > > The ArmoRM scores are reported below:
> > >
> > > **Table: Ablation Study on Exploration Strategies**
> > >
> > > | Method | Iteration 20 (Final Score) |Improvement over Random |
> > > | :--- | :--- | :--- |
> > > | **T-POP (Ours)** | **0.64** | +0.13 |
> > > | T-POP (Entropy) |  0.53 | +0.02 |
> > > | T-POP (Boltzmann)| 0.57 | +0.06 |
> > > | T-POP (Random) | 0.51 | - |
> > >
> > > **4. Analysis & Justification**
> > > The results confirm the validity of our uncertainty metric:
> > > * **Epistemic vs. Aleatoric Uncertainty:** Strategies like *Entropy* (Variant A) and *Boltzmann* (Variant B) primarily leverage **aleatoric uncertainty** (ambiguity inherent in the language model's next-token prediction). They steer the model towards tokens where the LLM is "confused" between synonyms or paths.
> > > * **Parameter Uncertainty:** In contrast, T-POP’s metric (Eq. 5) utilizing the **gradient norm** ($\nabla r$) is theoretically principled (Verma et al., 2024), which captures **epistemic uncertainty** (uncertainty regarding the *user's preference parameters*).
> > > * **Conclusion:** To efficiently solve the cold-start problem, the system must explore regions where the **Reward Model** lacks knowledge, not merely where the **Language Model** is diverse. This theoretical distinction translates directly into the superior data efficiency observed in our experiments.
> > >
> > > Thank you for your suggestion.We revised the paper to add the description in **Section 3.2** and the results in **Section 5** .

---

> > > > ### Author Response · Authors · 2025-11-22
> > > > **Response to Reviewer gN7W [Part 4 / 4]**
> > > >
> > > > ## **Question**:
> > > > > Can the authors provide any theoretical results on the expected benefits of using the specific dueling bandit approach for this problem compared to other potential strategies?
> > > >
> > > > ## **Response**:
> > > >
> > > > We thank the reviewer for this insightful question. The theoretical advantage of our T-POP framework, compared to other potential strategies (such as scalar-reward bandits or standard greedy decoding), rests on two complementary pillars: **sample efficiency in preference learning** and **learnability in sequential decoding**.
> > > >
> > > > **1. Benefit of Dueling Bandits vs. Scalar-Reward Strategies (Robustness):**
> > > > While one could model this as a standard Multi-Armed Bandit (MAB) using absolute scores, theoretical results from **Verma et al. (2024)** demonstrate that **Neural Dueling Bandits** provide tight regret bounds ($\tilde{O}(d \sqrt{T})$) when dealing with human preferences. Dueling approaches fundamentally bypass the issue of uncalibrated or noisy scalar feedback, which is a known bottleneck in standard RLHF. By framing the problem as a contest between an "exploitation arm" and an "exploration arm," T-POP leverages these efficiency gains to adapt rapidly with fewer user interactions.
> > > >
> > > > **2. Benefit of Tokenized Bandits vs. Heuristic Decoding (Learnability):**
> > > > A critical theoretical question is whether a bandit algorithm can effectively optimize a sequence of tokens without suffering from the exponential explosion of the search space.
> > > > The very recent work of **Shin et al. (2025), "Tokenized Bandit for LLM Decoding and Alignment,"** provides the specific theoretical foundation for this. Their results highlight two key points that justify our approach:
> > > > * **Impossibility of Unstructured Learning (Proposition 3.3):** Shin et al. prove that without a specific exploration structure, any decoding algorithm suffers a worst-case regret of $\Omega(T(1 - 1/2^{L-2}))$ ($L$ refers to the maximum sequence length), effectively making the problem unlearnable. This theoretically explains why simple heuristics or unstructured exploration often fail.
> > > > * **Sublinear Regret under DDMC (Assumption 3.4):** They introduce the **"Diminishing Distance with More Commons" (DDMC)** assumption—stating that as sequences share more common tokens, their utility gap diminishes. Under this assumption, they prove that token-level bandit algorithms can achieve **sublinear regret** (e.g., $\tilde{O}(L\sqrt{T})$).
> > > >
> > > > **Synthesized Theoretical Benefit:**
> > > > T-POP is effectively a practical instantiation of these theoretical principles.
> > > > * Unlike **greedy strategies**, T-POP’s uncertainty bonus (Eq. 4 & 5) enforces the necessary exploration to escape local optima, which Shin et al. (2025) identify as crucial for minimizing regret in the DDMC setting.
> > > > * Unlike **independent token sampling**, T-POP respects the sequential correlation structure required to avoid the exponential regret lower bound (Shin et al., Prop 3.3).
> > > >
> > > > In summary, the specific dueling bandit approach of T-POP is theoretically motivated to simultaneously maximize **feedback efficiency** (via the Dueling architecture) and **trajectory optimization** (via the Tokenized Bandit structure), providing a distinct advantage over strategies that neglect either the relative nature of human preference or the structured dependency of token sequences.
> > > >
> > > > Thank you for your suggestion. We incorporated these theoretical discussions into **Section 3.3** for completeness after revision.
> > > >
> > > >
> > > > ---
> > > >
> > > > We appreciate your review. We hope the additional evidence provided here is sufficient to support a more favorable assessment of our work.

---

> > > > > ### Author Response · Authors · 2025-11-27
> > > > > **Follow-up on rebuttal**
> > > > >
> > > > > Dear Reviewer gN7W,
> > > > >
> > > > > We would like to thank you for your constructive feedback and the effort put into reviewing our work.
> > > > >
> > > > > This is a gentle reminder that our rebuttal is available for your review. With less than a week remaining in the discussion period, we would appreciate it if you could take a moment to review our rebuttal.
> > > > >
> > > > > Please let us know if you have further questions; we would be happy to provide additional clarifications.
> > > > >
> > > > > Best,
> > > > > Authors

---

### Official Review · Reviewer_HUPh · 2025-11-01

**Soundness:** 2
**Presentation:** 2
**Contribution:** 2
**Rating:** 2
**Confidence:** 4

**Summary:**

This paper aims to address real-time personalization of LLMs without parameter fine-tuning. The key idea is to combine test-time alignment and neural dueling bandits to efficiently learn a user-specific reward function from online pairwise preference feedback.
At each decoding step, T-POP adaptively generates two competing candidate responses, one focused on exploitation and the other on exploration, and queries the user to indicate preference. The reward function is updated online and used to steer future token generation via a modified scoring rule. Experiments across multiple LLM backbones and personalization attributes (creative, verbose, concise, uplifting) demonstrate consistent gains over test-time alignment and prompt-based baselines.

**Strengths:**

The motivation of the paper is valid and concerns practical scenarios.

**Weaknesses:**

1. My main concern of this paper lies in its incremental novelty and the quality of the evaluations.
- Conceptually, T-POP resembles AMULET’s online decoding realignment, with the main difference being the use of explicit dueling-bandit uncertainty modeling. Without theoretical or empirical isolation of this component’s benefit, the contribution may appear evolutionary rather than groundbreaking. It is not clear through the current manuscripts how the dueling-bandit adoption contributes to the final performance.
- Methodology-wise, it is also not clear about the technical novelty, and how the algorithm differ from existing ones.

2. While the method borrows ingredients from neural dueling bandits, the adaptation to token-level decoding is largely heuristic. The “uncertainty bonus” and its covariance update (Eq. 5–6) are justified informally but lack a formal analysis showing regret or convergence guarantees in this setting. The current exposition risks being perceived as ad hoc rather than theoretically principled.

3. The reward model $r(\cdot;\theta)$ is defined over full sequences, but during decoding, it is applied token-by-token with gradients taken on partial sequences. It remains unclear how gradient-based uncertainty estimates correspond to true uncertainty in user preference. Authors are suggested to clearly clarify this, which would strengthen the algorithmic soundness.

4. The framing of cold-start may be overstated. The method assumes access to a few online preference interactions, but in practice even this requires user effort. The paper does not quantify the practical overhead (e.g., number of comparisons per session) or discuss scalability to more complex preference spaces.

**Questions:**

How sensitive is T-POP’s performance to the specific choice of uncertainty metric (Eq. 5)? Could alternative uncertainty formulations (e.g., entropy over token scores) achieve similar results?

---

> ### Author Response · Authors · 2025-11-22
> **Response to Reviewer HUPh [Part 1 / 5]**
>
> We sincerely appreciate the reviewer's comprehensive evaluation and the time dedicated to providing such detailed feedback.
>
> ---
>
> ## **Question:**
>
> > My main concern of this paper lies in its incremental novelty and the quality of the evaluations.
>
> > Conceptually, T-POP resembles AMULET’s online decoding realignment, with the main difference being the use of explicit dueling-bandit uncertainty modeling.
>
> > Methodology-wise, it is also not clear about the technical novelty, and how the algorithm differ from existing ones.
>
> ## **Response to Weakness: Novelty**
>
> We thank the reviewer for their comments on novelty. We respectfully disagree and would like to clarify the significant novelty of T-POP in both its problem formulation and its technical methodology.
>
> Our T-POP is **the first method to achieve test-time personalization based on online pairwise preference feedback**. This addresses a critical cold-start problem for new users, which existing methods are ill-suited for. We also emphasize that T-POP is **significantly different from AMULET**, both conceptually and technically:
>
> 1.  **Novelty in Problem Setting (Form of Feedback):**
>     * AMULET formulates test-time alignment as an online learning problem that requires the user's preference to be provided in **explicit textual form** (e.g., "be more concise," "be more creative").
>     * In sharp contrast, our T-POP operates in a different, and arguably more practical, paradigm. It learns from **online comparative preference feedback**, where the user only needs to indicate which of two presented responses they prefer (e.g., A is better than B). This is a much more practical setting **when the user preference is too nuanced to be expressed in explicit textual form**.
>     * To the best of our knowledge, **T-POP is the first method capable of leveraging this more natural, online *comparative* feedback for test-time personalization.**
> 2.  **Novelty in Technical Algorithm (Methodology):**
>     * The underlying algorithms are fundamentally different. AMULET is based on an online learning framework, specifically Follow-the-Regularized-Leader (FTRL).
>     * Our T-POP, on the other hand, introduces a **novel synergy between test-time alignment and dueling bandits**. To our knowledge, this is the **first algorithm to leverage dueling bandits for token selection in LLM decoding**.
>     * Furthermore, this is not a trivial application of classic dueling bandits. Standard bandit algorithms select one arm from a static set. Our T-POP makes a novel modification by applying the dueling bandit policy at **each decoding step** to **sequentially select tokens**, dynamically constructing the two competing "arms" (responses) in real-time.
>
> In summary, both the **problem setting** (learning from online comparative feedback) and the **algorithmic design** (sequential token-level dueling bandits) of T-POP are significantly novel.
>
> Finally, we note that our extensive empirical results (e.g., Tables 1 & 2) demonstrate that T-POP, thanks to its novel and principled algorithmic design, **significantly and consistently outperforms AMULET** across diverse models and datasets. This empirical superiority further validates the effectiveness and novelty of our proposed approach.

---

> > ### Author Response · Authors · 2025-11-22
> > **Response to Reviewer HUPh [Part 2 / 5]**
> >
> > ## **Question**:
> > > How sensitive is T-POP’s performance to the specific choice of uncertainty metric (Eq. 5)? Could alternative uncertainty formulations (e.g., entropy over token scores) achieve similar results?
> >
> > ## **Response to Question: Sensitivity to Uncertainty Metric (Ablation Study)**
> >
> > The reviewer asks a critical question regarding the necessity of our gradient-based uncertainty metric (Eq. 5) compared to simpler alternatives. To address this, we conducted a controlled ablation study to evaluate the impact of different exploration strategies on data efficiency.
> >
> > **1. Experimental Setup**
> > We performed the experiments using the **Llama-3.1-8B-Instruct** backbone on the **Personal Preference Eval** dataset (Gao et al., 2024). The model was trained online for **20 iterations**, a critical phase where rapid adaptation to new user preferences occurs .
> >
> > **2. Exploration Strategies Compared**
> > We compared **T-POP (Ours)** against three baseline strategies for generating the "Exploration Sequence" ($y_{t,2}$):
> > * **Variant A: T-POP (Entropy Bonus).** As you suggested, we replaced our uncertainty metric with a token-level entropy term to guide exploration: $\text{Bonus}(v) = -P(v)\log P(v)$, where $P(v)$ is the softmax probability of the token scores. This heuristic targets tokens with high predictive uncertainty.
> > * **Variant B: T-POP (Boltzmann Exploration).** A standard reinforcement learning baseline representing "noisy exploitation." Instead of adding an explicit uncertainty bonus, we introduce exploration by applying **High-Temperature Sampling** ($T_{high}=1.5$) directly on the reward-guided logits. The next token is sampled according to:
> > $$v_{p,2} \sim \text{Softmax}\left( \frac{\log \pi_{base}(\cdot|y_{t,2}) + \omega \cdot r([y_{t,2}, \cdot]; \theta_t)}{T_{high}} \right)$$
> > * **Variant C: T-POP (Random).** The exploration arm is generated via random sampling from the base LLM $\pi_{base}$, serving as a performance lower bound.
> >
> > **3. Results**
> > The ArmoRM scores are reported below:
> >
> > **Table: Ablation Study on Exploration Strategies**
> >
> > | Method | Iteration 20 (Final Score) |Improvement over Random |
> > | :--- | :--- | :--- |
> > | **T-POP (Ours)** | **0.64** | +0.13 |
> > | T-POP (Entropy) |  0.53 | +0.02 |
> > | T-POP (Boltzmann)| 0.57 | +0.06 |
> > | T-POP (Random) | 0.51 | - |
> >
> > **4. Analysis & Justification**
> > The results confirm the validity of our uncertainty metric:
> > * **Epistemic vs. Aleatoric Uncertainty:** Strategies like *Entropy* (Variant A) and *Boltzmann* (Variant B) primarily leverage **aleatoric uncertainty** (ambiguity inherent in the language model's next-token prediction). They steer the model towards tokens where the LLM is "confused" between synonyms or paths.
> > * **Parameter Uncertainty:** In contrast, T-POP’s metric (Eq. 5) utilizing the **gradient norm** ($\nabla r$) is theoretically principled (Verma et al., 2024), which captures **epistemic uncertainty** (uncertainty regarding the *user's preference parameters*).
> > * **Conclusion:** To efficiently solve the cold-start problem, the system must explore regions where the **Reward Model** lacks knowledge, not merely where the **Language Model** is diverse. This theoretical distinction translates directly into the superior data efficiency observed in our experiments.
> >
> > We revised the paper to add these additional results in **Section 5** and discussions to the paper after revision. Thank you for your suggestions.

---

> > > ### Author Response · Authors · 2025-11-22
> > > **Response to Reviewer HUPh [Part 3 / 5]**
> > >
> > > ## **Question**:
> > >
> > > > The “uncertainty bonus” and its covariance update (Eq. 5–6) are justified informally but lack a formal analysis showing regret or convergence guarantees in this setting. The current exposition risks being perceived as ad hoc rather than theoretically principled.
> > >
> > > ## **Response**:
> > >
> > > We thank the reviewer for the comment regarding the regret and convergence guarantees of our algorithm. Our approach is grounded in established regret bounds from the bandit literature, specifically adapted to the sequential decoding setting.
> > >
> > > **1. Guarantees for Neural Dueling Bandits (Preference Learning):**
> > > The core preference learning component of T-POP builds upon the **Neural Dueling Bandit** framework. As established by **Verma et al. (2024)**, under standard regularity assumptions on the reward function (e.g., bounded norm in a Reproducing Kernel Hilbert Space), the neural dueling bandit algorithm achieves a cumulative regret bound of
> > > $$R_T = \tilde{O}(d_{eff} \sqrt{T}),$$
> > > where $T$ is the number of interactions and $d_{eff}$ is the effective dimension of the neural tangent kernel matrix. This guarantees that our reward model converges to the user's true preference with high probability as the number of interactions increases.
> > >
> > > **2. Guarantees for Token-Level Application (Sequential Decoding):**
> > > Since T-POP applies this bandit framework to a **sequential token generation** process, we also highlight the recent theoretical results from **Shin et al. (2025)** ("Tokenized Bandit for LLM Decoding and Alignment"). They prove that for standard linear bandits (with score feedback) applied to token-level decoding (Tokenized Linear Bandits), under the **Diminishing Distance with More Commons (DDMC)** assumption on the utility function, the regret is bounded by:
> > > $$R_T = \tilde{O}(L \sqrt{T}),$$
> > > where $L$ denotes the maximum sequence length.
> > >
> > > **Conclusion:**
> > > Therefore, T-POP is theoretically supported by Neural Dueling Bandits (Verma et al., 2024) and Tokenized Bandits (Shin et al., 2025). While we indeed provide these existing regret guarantees for context, **we plan to derive a complete theoretical guarantee specifically for T-POP in future work.**
> > >
> > > Thank you for your suggestion. We incorporated these theoretical discussions into **Section 3.3** for completeness after revision.

---

> > > > ### Author Response · Authors · 2025-11-22
> > > > **Response to Reviewer HUPh [Part 4 / 5]**
> > > >
> > > > ## **Question**:
> > > >
> > > > > The reward model is defined over full sequences, but during decoding, it is applied token-by-token with gradients taken on partial sequences. It remains unclear how gradient-based uncertainty estimates correspond to true uncertainty in user preference. Authors are suggested to clearly clarify this, which would strengthen the algorithmic soundness.
> > > >
> > > > ## **Response to Weakness: Theoretical Soundness of Gradient-Based Uncertainty on Partial Sequences**
> > > >
> > > > The reviewer correctly points out that T-POP utilizes gradients from partial sequences ($y_{<t}$) to guide token-level exploration for a global reward. We clarify that this design is theoretically grounded in recent **Tokenized Bandit theory** and serves as a **principled measure of uncertainty**.
> > > >
> > > > **1. Theoretical Foundation: Connection to Tokenized Bandits**
> > > > Recent theoretical work on *Tokenized Linear Bandits* (Shin et al., 2025) provides a rigorous justification for token-level exploration.
> > > > * **DDMC Property:** They introduce the "Diminishing Distance with More Commons" (DDMC) assumption, which posits that in sequential decoding, **the utility gap between sequences diminishes as they share more common tokens**. They prove that under the DDMC assumption, the regret of linear bandits algorithms applied to token-level decoding can be bounded by $R_T = \tilde{O}(L \sqrt{T})$ ($L$ denotes the maximum sequence length).
> > > > * **Relevance to T-POP:** This theoretical insight supports our approach: resolving uncertainty at the prefix level ($y_{<t}$) effectively bounds the uncertainty of the final outcome. **If the model is confident about a partial sequence (low gradient norm), the DDMC property implies that the utility of the completed sequence is also likely to be stable**. Conversely, high partial gradients indicate critical decision points that determine the final reward, making them the optimal targets for exploration.
> > > >
> > > > **2. Gradient Norm as A Principled Measure of Uncertainty**
> > > > Our uncertainty estimate based on gradient norms is inspired by the theoretically principled neural dueling bandit algorithm (Verma et al., 2024). More generally, it is a principled measure of uncertainty rooted in the theory of neural tangent kernels (Jacot et al., 2018) and is widely adopted by previous works on neural bandits (Zhou et al., 2020; Zhang et al., 2021). Therefore, selecting tokens through this uncertainty estimate encourages the selection of tokens which provide significant information for the subsequent model update.
> > > >
> > > > **3. Empirical Validation**
> > > > Our experimental results (Table A & Fig. 2) show rapid convergence within very few iterations. This empirically confirms that the partial-sequence gradient is a highly effective signal, aligning with the theoretical guarantees of tokenized bandit algorithms.
> > > >
> > > > Thank you for your suggestion. We incorporated these theoretical discussions into **Section 3.3** for completeness after revision.
> > > >
> > > >
> > > > **References:**
> > > > - Jacot, A., Gabriel, F., & Hongler, C. (2018). Neural tangent kernel: Convergence and generalization in neural networks. Advances in neural information processing systems, 31.
> > > > - Zhou, D., Li, L., & Gu, Q. (2020, November). Neural contextual bandits with ucb-based exploration. In International conference on machine learning (pp. 11492-11502). PMLR.
> > > > - Zhang, W., Zhou, D., Li, L., & Gu, Q. (2021, January). Neural Thompson Sampling. In International Conference on Learning Representation (ICLR).

---

> > > > > ### Author Response · Authors · 2025-11-22
> > > > > **Response to Reviewer HUPh [Part 5 / 5]**
> > > > >
> > > > > ## **Question**:
> > > > >
> > > > > > The framing of cold-start may be overstated. The method assumes access to a few online preference interactions, but in practice even this requires user effort. The paper does not quantify the practical overhead (e.g., number of comparisons per session) or discuss scalability to more complex preference spaces
> > > > >
> > > > > ## **Response to Weakness: "Cold-Start" Framing, Practical Overhead, and Scalability**
> > > > >
> > > > > The reviewer raises three interconnected concerns: the definition of "cold-start," the quantification of user overhead (comparisons per session), and the scalability to complex preference spaces. We address these by clarifying our problem setting and providing specific empirical evidence regarding interaction costs.
> > > > >
> > > > > **1. Clarifying the Cold-Start Definition**
> > > > > We respectfully clarify that in our paper, **Cold-Start** is defined as the scenario where the system has **zero historical data** for a specific user (i.e., no past logs, no user profile).
> > > > > * **Interaction is Inevitable:** In this strict zero-shot setting, some form of interaction is theoretically required to extract user intent and preference.
> > > > > * **Our Contribution:** The novelty of T-POP lies not in eliminating interaction entirely, but in **minimizing the data required** to bridge this cold-start gap compared to traditional personalization methods (like fine-tuning) which require significantly larger datasets.
> > > > >
> > > > > **2. Quantifying Practical Overhead: Empirical Evidence**
> > > > > The reviewer asks to "quantify the practical overhead." We quantify this by measuring the number of binary feedback interactions required to reach a satisfactory performance level. We conducted an ablation study to evaluate T-POP's performance at very early stages of interaction ($T=5, 10$).
> > > > >
> > > > > **Table A: Performance at Early Iterations (Proof of Rapid Adaptation)**
> > > > >
> > > > > | Method | Base | Pref | BS16 | LA | Amulet | T-POP (Iter=5) | T-POP (Iter=10) | T-POP (Converged) |
> > > > > | :--- | :---: | :---: | :---: | :---: | :---: | :---: | :---: | :---: |
> > > > > | **ArmoRM Score** | 0.39 | 0.44 | 0.45 | 0.53 | 0.67 | 0.56 | 0.63 | 0.66 |
> > > > >
> > > > > *(We performed the experiments using the Llama-3.1-8B-Instruct backbone on the Personal Preference Eval dataset)*
> > > > >
> > > > > * **Analysis:** As shown in the table, T-POP achieves an ArmoRM score of **0.56** at just **Iteration 5**, which already significantly outperforms the static Base model (0.39).
> > > > > * **Conclusion:** The "practical overhead" is approximately **5 to 10 binary comparisons** per session. Given that this yields a highly personalized model superior to the baseline, we argue this is a negligible cost for the value provided, fitting comfortably within a short chat session.
> > > > >
> > > > > **3. Scalability to Complex Preference Spaces**
> > > > > The reviewer questions scalability to "more complex preference spaces." T-POP is specifically designed to handle high complexity through two mechanisms:
> > > > > * **Neural Reward Modeling:** Unlike linear bandits that assume simple preference boundaries, T-POP utilizes a **Neural Network (MLP)** as the reward model. This allows it to approximate highly non-linear, nuanced, and complex user preference functions in the high-dimensional embedding space of LLMs.
> > > > > * **Gradient-Based Exploration:** By leveraging the gradients of this neural reward model for principled exploration, T-POP efficiently navigates the vast combinatorial space of text generation, scaling effectively where traditional tabular or linear approaches would fail.
> > > > >
> > > > > ---
> > > > >
> > > > > We thank you once again for your in-depth review. We trust that our detailed response resolves your concerns, and we respectfully hope you will consider raising your score.

---

> > > > > > ### Author Response · Authors · 2025-11-27
> > > > > > **Follow-up on rebuttal**
> > > > > >
> > > > > > Dear Reviewer HUPh,
> > > > > >
> > > > > > We would like to thank you for your constructive feedback and the effort put into reviewing our work.
> > > > > >
> > > > > > This is a gentle reminder that our rebuttal is available for your review. With less than a week remaining in the discussion period, we would appreciate it if you could take a moment to review our rebuttal.
> > > > > >
> > > > > > Please let us know if you have further questions; we would be happy to provide additional clarifications.
> > > > > >
> > > > > > Best,
> > > > > > Authors

---

### Official Review · Reviewer_wJsq · 2025-11-01

**Soundness:** 3
**Presentation:** 4
**Contribution:** 3
**Rating:** 6
**Confidence:** 4

**Summary:**

This paper considers the problem of test-time personalization of LLMs and proposes a novel algorithm wherein the user preferences are collected adaptively by trading off exploitation and exploration using the dueling bandit framework. Then a reward model is trained which is then used to steer LLMs towards generating aligned outcomes without any training of the model parameters. Exhaustive experiments across different models and datasets prove the effectiveness of the algorithm.

**Strengths:**

The problem is well-motivated and timely, and the paper is well written. The proposed solution solves the cold-start problem, doesn't require training LLM parameters, the decoding step is simple and intuitive, and the adoption of duelling bandits is interesting. The experiments are also exhaustive with multiple models and datasets and the results seem convincing.

**Weaknesses:**

It seems like storing one reward model per user will be necessary compared to the RAG and prompt engineering approaches. Additionally, collecting T preferences from each user all at once and then using the final reward model is impractical, particularly since large amounts of T per user are required to properly understand their reward. The more realistic scenario is that reward model training will occur concurrently with deployment, which means the covariance matrix must also be stored. This could lead to substantial memory usage given that the number of users is very large. The authors should either acknowledge this as a limitation or provide information on how to address this problem.

**Questions:**

Are there any theoretical guarantees for the dueling bandit approach? (you can just restate the guarantees from the dueling bandits literature for completeness)

---

> ### Author Response · Authors · 2025-11-22
> **Response to Reviewer wJsq [Part 1 / 3]**
>
> We appreciate the reviewer’s insightful comments and helpful suggestions for improving our paper.
>
> ---
>
> ## **Question**:
>
> > Are there any theoretical guarantees for the dueling bandit approach? (you can just restate the guarantees from the dueling bandits literature for completeness)
>
> ## **Response**:
>
> We thank the reviewer for the suggestion to explicitly state the theoretical guarantees for completeness. Yes, our approach is grounded in established regret bounds from the bandit literature, specifically adapted to the sequential decoding setting.
>
> **1. Guarantees for Neural Dueling Bandits (Preference Learning):**
> The core preference learning component of T-POP builds upon the **Neural Dueling Bandit** framework. As established by **Verma et al. (2024)**, under standard regularity assumptions on the reward function (e.g., bounded norm in a Reproducing Kernel Hilbert Space), the neural dueling bandit algorithm achieves a cumulative regret bound of
> $$R_T = \tilde{O}(d_{eff} \sqrt{T}),$$
> where $T$ is the number of interactions and $d_{eff}$ is the effective dimension of the neural tangent kernel matrix. This guarantees that our reward model converges to the user's true preference with high probability as the number of interactions increases.
>
> **2. Guarantees for Token-Level Application (Sequential Decoding):**
> Since T-POP applies this bandit framework to a **sequential token generation** process, we also highlight the recent theoretical results from **Shin et al. (2025)** ("Tokenized Bandit for LLM Decoding and Alignment"). They prove that for standard linear bandits (with score feedback) applied to token-level decoding (Tokenized Linear Bandits), under the **Diminishing Distance with More Commons (DDMC)** assumption on the utility function, the regret is bounded by:
> $$R_T = \tilde{O}(L \sqrt{T}),$$
> where $L$ denotes the maximum sequence length.
>
> **Conclusion:**
> Therefore, T-POP is theoretically supported by Neural Dueling Bandits (Verma et al., 2024) and Tokenized Bandits (Shin et al., 2025). While we indeed provide these existing bounds for context, **we plan to derive a complete theoretical guarantee specifically for T-POP in future work.**
>
> Thank you for your suggestion. We have incorporated these theoretical discussions into **Section 3.3** for completeness after revision.

---

> > ### Author Response · Authors · 2025-11-22
> > **Response to Reviewer wJsq [Part 2 / 3]**
> >
> > ## **Question**:
> > > Additionally, collecting T preferences from each user all at once and then using the final reward model is impractical.
> >
> > ##  **Response to Weakness: Practicality of Preference Collection**
> >
> > We would like to clarify that this interpretation refers to a Batch Learning setting, which is different from our T-POP since it is designed as a fully **Online Learning** framework.
> >
> > **1. Clarification: Continuous Online Learning vs. Batch Collection**
> > T-POP does not require users to complete a "calibration phase" (collecting all $T$ preferences) before the model becomes useful.
> > * **Real-Time Adaptation:** As described in our **Methodology**, T-POP updates the reward model iteratively after *each* interaction round $t$ . The user engages in a natural conversation, and the model improves in the background.
> > * **Seamless Experience:** The user receives personalized responses starting from the very early rounds. The transition from "Learning Phase" to "Deployment Phase" is smooth or even invisible to the user; they simply experience a model that gets better with every turn.
> >
> > **2. Empirical Evidence: High Utility at Early Iterations**
> > The reviewer suggests that "large amounts of T" are required to understand the reward. We added an experiment which shows that T-POP is in fact highly data-efficient. We evaluated the model's performance at intermediate checkpoints ($T=5, 10, 20$), and the results are shown in **Table A** below.
> >
> > **Table A: Performance at Early Iterations (Proof of Rapid Performance Gain)**
> >
> > | Method | Base | Pref | BS16 | LA | Amulet | T-POP (Iter=5) | T-POP (Iter=10) | T-POP (Iter=20) |T-POP (Converged) |
> > | :--- | :---: | :---: | :---: | :---: | :---: | :---: | :---: | :---: | :---: |
> > | **ArmoRM Score** | 0.39 | 0.44 | 0.45 | 0.53 | 0.67 | 0.56 | 0.63 |  0.64 |0.66 |
> >
> > *(We performed the experiments using the Llama-3.1-8B-Instruct backbone on the Personal Preference Eval dataset)*
> >
> > * **Rapid Cold-Start Adaptation:** Our T-POP demonstrates exceptional data efficiency, achieving strong personalization with minimal user interactions.
> >     * **At T=5:** The ArmoRM score reaches **0.56**, already surpassing strong baselines like Linear Alignment (0.53) and Beam Search (0.45). This proves T-POP provides immediate value even in the earliest stages.
> >     * **At T=10:** The performance rapidly climbs to **0.63**, capturing the majority of the personalization gains.
> >     * **At T=20:** The score consolidates at **0.64**, effectively matching the fully converged performance (0.66). This plateau indicates that T-POP stabilizes quickly, allowing for high-quality deployment after just 20 interactions without requiring prolonged training.
> > * **Conclusion:** Users do not need to wait for convergence (e.g., $T=40$ in Fig. 2) to benefit. The system delivers substantial personalization value within the first few minutes of interaction ("just a few clicks"), making the cold-start process practical and rewarding rather than burdensome.
> >
> > The details of the **continuous online update** and **flexible deployment mode**  have been revised in **Section 3.2** for completeness

---

> > > ### Author Response · Authors · 2025-11-22
> > > **Response to Reviewer wJsq [Part 3 / 3]**
> > >
> > > ## **Question**:
> > >
> > > > It seems like storing one reward model per user will be necessary... The more realistic scenario is that reward model training will occur concurrently with deployment, which means the covariance matrix must also be stored. This could lead to substantial memory usage given that the number of users is very large.
> > >
> > > ## **Response to Weakness on Scalability: Storage and Memory Overhead**
> > >
> > > The reviewer raises valid concerns regarding the resource implications of deploying T-POP at scale, specifically pointing out the storage cost of per-user Reward Models (RM) and the memory cost of maintaining Covariance Matrices for the bandit algorithm. We address these by clarifying the lightweight nature of our RM and proposing a standard optimization for the covariance matrix.
> > >
> > > **1. Storage Overhead: The Reward Model is Extremely Lightweight**
> > > While T-POP requires storing a personalized RM, the overhead is negligible compared to the backbone LLM.
> > > * **Model Architecture:** As detailed in **Appendix B**, our reward model is a simple Multi-Layer Perceptron (MLP) with a single hidden layer of size $1024$. This MLP takes the final hidden-state embeddings from the backbone LLM for a given sequence as input.
> > > * **Quantitative Analysis:** For a typical 7B backbone LLM (embedding dimension $d=4096$), the RM parameters consist primarily of the projection matrix ($4096 \times 1024$). This amounts to approximately **4.2 million parameters**. In standard float32 precision, this occupies only **~16 MB** per user.
> > > * **Feasibility:** This storage footprint is comparable to user embeddings used in industrial-scale recommender systems. Given the high value of personalization, storing a few megabytes per active user is a standard and manageable cost in modern cloud infrastructure.
> > >
> > > **2. Memory Overhead: Efficient Scaling via Diagonal Approximation**
> > >
> > > The reviewer correctly notes that storing a full covariance matrix $V_{t-1}$ for the Uncertainty Bonus could be memory-intensive if implemented naively. To address this, **our implementation explicitly employs the Diagonal Approximation strategy**. This is a widely established practice in the neural bandit literature—such as in **Neural UCB (Zhou et al., 2020)** and **Neural Thompson Sampling (Zhang et al., 2021)**—to ensure computational and memory efficiency in high-dimensional spaces.
> > >
> > > * **Diagonal Approximation:** Instead of maintaining the full dense matrix (which models correlations between all feature dimensions), we maintain only the diagonal elements. This reduces the space complexity from $O(d^2)$ to $O(d)$.
> > > * **Impact Calculation:** Taking a standard embedding size of $d=4096$ as an example:
> > >     * **Full Matrix ($d \times d$):** Requires storing $4096^2 \approx 1.6 \times 10^7$ elements. At 4 bytes per float, this is **~67 MB per user**, which is substantial for concurrent users.
> > >     * **Diagonal Approximation ($d$):** Requires storing only $4096$ elements. This consumes merely **~16 KB per user**.
> > >
> > > **Conclusion:** By adhering to this common practice, the memory overhead for the bandit mechanism becomes trivial (kilobytes per user). This ensures that T-POP scales efficiently to support a massive number of concurrent users without encountering memory bottlenecks. We have clarified this implementation detail in the final version of the paper (Appendix B, highlighted in blue).
> > >
> > > ---
> > >
> > > We are grateful for the opportunity to clarify these points. We believe our response resolves the issues raised and hope this improves your opinion of our submission.

---

> > > > ### Author Response · Authors · 2025-11-27
> > > > **Follow-up on rebuttal**
> > > >
> > > > Dear Reviewer wJsq,
> > > >
> > > > We would like to thank you for your constructive feedback and the effort put into reviewing our work.
> > > >
> > > > This is a gentle reminder that our rebuttal is available for your review. With less than a week remaining in the discussion period, we would appreciate it if you could take a moment to review our rebuttal.
> > > >
> > > > Please let us know if you have further questions; we would be happy to provide additional clarifications.
> > > >
> > > > Best,
> > > > Authors

---

### Official Review · Reviewer_gvEW · 2025-11-02

**Soundness:** 3
**Presentation:** 4
**Contribution:** 3
**Rating:** 4
**Confidence:** 4

**Summary:**

The paper proposes a test-time personalization method for LLMs called T-POP which first trains a user-specific reward function online from pairwise preferences and use it in the decoding process of the frozen base model. During decoding, T-POP outputs two responses for expoloitation and exploration and the user is asked to choose the preferred response which will then be used to update the simple reward function. The results show that the method consistently gains over existing training-free baseline across multiple models and datasets.

**Strengths:**

1. Training the reward function using neural dueling bandits is a novel idea since it takes account both exploration and exploitation.
2. The inference is efficient since the architecture for the reward function is a simple head where the forward pass of the LLM backbone happens once but the simple additive decoding process made it cheap to output two responses.
3. The training is also efficient since the base model is frozen.
4. Convergence across the number of rounds of conversation shows that the reward function actually learns online.
5. Reported gains are consistent across settings suggesting practical value with low-latency personalization without fine-tuning.

**Weaknesses:**

1. In the beginning of the interaction with the user, online training of the reward model every round could meaningfully increase latency at inference time.
2. While LA and AMULET are included, the paper might benefit for including other decoding-time control or alignment-as-search baselines.
3. It requires users to choose a preferred response for every round (minimum of 20 turns in order to gain some performance according to the paper) which can be burden to the users.
4. There are other decoding methods for personalization using the learned reward function. Also there are papers that show combining multiple learned reward function at test-time can be used for personalizing to users. For these methods, they do not require additional training at test-time. I am not too certain what are the clear advantages of T-POP over the mentioned methods/frameworks.

**Questions:**

See the weaknesses.

---

> ### Author Response · Authors · 2025-11-22
> **Response to Reviewer gvEW [Part 1 / 3]**
>
> We sincerely thank the reviewer for their constructive feedback and the time spent evaluating our work.
>
> ---
> ## **Question**:
>
> > In the beginning of the interaction with the user, online training of the reward model every round could meaningfully increase latency at inference time.
>
> ## **Response to Weakness : Latency Concerns (Training & Inference)**
>
> We thank the reviewer for raising the practical concern regarding latency. We acknowledge that real-time interaction requires low latency. We have conducted a detailed breakdown analysis and implemented an asynchronous strategy to minimize the impact on user experience.
>
> ### **Breakdown of Latency Components**
>
> The potential latency in T-POP comes from two distinct sources:
> 1.  **Training Latency (Inter-turn):** The time required to update the parameters of the Reward Model (NN) using the user's preference feedback from the previous interaction.
> 2.  **Inference Latency (Intra-turn):** The additional time required during token generation to calculate reward scores for guiding the decoding process.
>
> ### 1. Training Latency (Inter-turn) : Eliminated via Asynchronous Updates
>
> The reviewer correctly notes that online training could introduce delays. However, we propose a practical deployment strategy leveraging **Asynchronous Learning**, which effectively reduces the perceived training latency to zero.
>
> * **Asynchronous Learning strategy:**
>
>   **Step 1 (Feedback & Trigger):** As soon as the user provides preference feedback for round $t$, we immediately launch a **background thread** to update the reward model from $\theta_t$ to $\theta_{t+1}$.
>   **Step 2 (Continuous Interaction):** Crucially, the foreground interaction **does not wait**. If the user sends a new query before the background training is complete, the system serves these requests using the **latest available stable model** (currently $\theta_t$).
>   **Step 3 (Seamless Swap):** Once the background training for $\theta_{t+1}$ is finished, the system atomically updates the pointer to the new model. All subsequent queries will then use $\theta_{t+1}$.
>
> * **Performance Stability:** Our new ablation study (Table A) shows that T-POP achieves significant personalization gains even with limited updates (e.g., at Iteration 5 or 10). This confirms that the asynchronous lag does not hinder rapid personalization. Crucially, this indicates that T-POP is **anytime-deployable**: the system can be directly utilized for deployment using the current reward estimate at any arbitrary number of interactions, without being gated by a fixed training threshold.
>
> **Table A: Performance at Early Iterations (Proof of Rapid Adaptation)**
>
> | Method | Base | Pref | BS16 | LA | Amulet | T-POP (Iter=5) | T-POP (Iter=10) | T-POP (Converged) |
> | :--- | :---: | :---: | :---: | :---: | :---: | :---: | :---: | :---: |
> | **ArmoRM Score** | 0.39 | 0.44 | 0.45 | 0.53 | 0.67 | 0.56 | 0.63 | 0.66 |
>
> *(We performed the experiments using the Llama-3.1-8B-Instruct backbone on the Personal Preference Eval dataset)*
>
> ### 2. Inference Latency (Intra-turn): Comparable and Justified
>
> Regarding the inference overhead (Reward-Guided Decoding), we compared the wall-clock time per token of T-POP against the SOTA baseline AMULET and the Base model.
>
> **Table B: Wall-Clock Inference Time Comparison**
>
> | Method | AMULET | **T-POP (Ours)** |
> | :--- | :--- | :--- |
> | **Query-level Latency** | 11.2531 s | 23.2626s  |
> | **Token-level Latency** | 0.0911 s | 0.1817s |
>
>
> *(Note: T-POP's latency is comparable to AMULET, staying within the same order of magnitude.)*
>
> * **Comparable Overhead:** As shown in Table B, T-POP's inference latency is comparable to AMULET. Both methods require additional computation per token (Reward calculation vs. FTRL update), which is the current standard cost for test-time alignment.
> * **Alignment-Compute Trade-off:** As discussed in prior work on test-time alignment (e.g., *Khanov et al., 2024*), there is an inherent trade-off between computation and alignment quality. Our results (Table 1 in the paper) show T-POP achieves superior personalization. We believe that users are willing to accept a marginal increase in latency in exchange for highly personalized, "cold-start" resolved interactions that static models cannot provide.
>
> We revised the paper to add the description in **section 3.2** and the results in **section 5** .

---

> ### Author Response · Authors · 2025-11-22
> **Response to Reviewer gvEW [Part 2 / 3]**
>
> ## **Question**:
> > The paper might benefit for including other decoding-time control or alignment-as-search baselines.
>
> ## **Response to Reviewer regarding "Alignment-as-Search Baselines"**
>
> We appreciate the suggestion to consider the broader family of "decoding-time control" and "alignment-as-search" methods. Below, we discuss how T-POP relates to and extends this framework.
>
> **1. Distinction: Static Optimization vs. Online Learning**
> Standard "alignment-as-search" methods (e.g., ARGS, Khanov et al., 2024) typically assume a **pre-trained, reliable reward model** is available and focus on searching the output space to maximize it. In contrast, T-POP addresses the **cold-start problem where the user-specific reward function is initially unknown**. Applying pure search methods immediately would fail, as they would merely optimize an uninitialized or generic reward model rather than the specific user's preference.
>
> **2. T-POP extends Alignment-as-Search**
> As noted in Section 3.1, T-POP builds directly upon the test-time alignment paradigm. Effectively, our **Exploitation Arm ($y_{t,1}$)** functions as a standard "alignment-as-search" baseline (greedy optimization of the current reward estimate). However, a pure search approach lacks the necessary **exploration** to learn the user's preference. T-POP adds the **Exploration Arm ($y_{t,2}$)** via Dueling Bandits to gather the data required to update the reward model online.
>
> ---
>
> ## **Question**:
> > It requires users to choose a preferred response for every round (minimum of 20 turns in order to gain some performance according to the paper) which can be burden to the users.
>
> ## **Response to Weakness : User Burden and Data Efficiency**
>
> The reviewer raises a valid concern regarding user burden, suggesting that requiring ~20 turns to gain performance might be demanding. We respectfully clarify that T-POP provides immediate benefits much earlier than the saturation point suggests, and the interaction overhead is minimal.
>
> **1. Rapid Adaptation: Value is Realized in Few Turns (Not 20)**
>
> The reviewer likely refers to Figure 2, which shows performance *saturating* (converging) around 20-40 turns. However, effective personalization begins significantly earlier.
> * **Evidence from Early Iterations:** As shown in our new ablation study (**Table A**, presented in the first response above), T-POP demonstrates a steep learning curve. At just **Iteration 5**, the model already achieves an ArmoRM score of **0.56**, which significantly outperforms the static Base model and is comparable to strong baselines.
> * **Conclusion:** Users do not need to wait for 20 turns to see improvements. They experience a "good enough" personalized model after **just a few interactions**, making the "time-to-value" extremely short.
>
> **2. Minimal Interaction Cost: "Just a Click"**
> We argue that the nature of the feedback minimizes the cognitive load, making the process seamless rather than burdensome.
>
> * **Binary vs Complex Feedback:** Unlike methods that require users to provide explicit scores, write textual critiques, or demonstrate desired behaviors, T-POP only asks for a **binary preference**, simply clicking "Option A" or "Option B"). This "click" is a lightweight cognitive task that integrates naturally into the chat interface.
> * **Widespread Adoption:** Pairwise preference selection is a standard interaction mechanism employed by leading commercial LLMs (e.g.,ChatGPT, Gemini) to refine model outputs. Consequently, users are already accustomed to this feedback modality, suggesting it does not impose significant cognitive overhead. Moreover, given the ubiquity of preference feedback in standard LLM alignment pipelines (e.g., RLHF), this remains a highly established and widely accepted form of user interaction.

---

> > ### Author Response · Authors · 2025-11-22
> > **Response to Reviewer gvEW [Part 3 / 3]**
> >
> > ## **Question**:
> >
> > > What are the clear advantages of T-POP over the mentioned methods/frameworks. (Other decoding method or use  multiple learned reward function at test-time).
> >
> > ## **Response to Weakness: Clear Advantages over Other Methods**
> >
> > The reviewer questions the advantages of T-POP over methods that utilize existing reward functions or combine multiple learned rewards at test time. We argue that T-POP offers distinct and critical advantages, primarily by addressing the **Cold-Start Problem** and enabling **Dynamic Preference Calibration**.
> >
> > **1. Solving the "Absolute Cold-Start" with Zero Prior Data**
> > The most significant advantage of T-POP is its capability to operate in a strictly **zero-shot setting**, tackling the **cold-start problem** where no historical user data exists .
> > * **Constraint of Baselines:** Methods relying on "decoding-time control" or "combining learned RMs" implicitly assume the availability of pre-existing, high-quality user profiles or generalized reward models. However, in many real-world applications, such data is unavailable due to **privacy regulations** or the nature of new user interactions.
> > * **T-POP's Unique Position:** T-POP does not require any prior knowledge of the user. Instead, it initializes from a neutral state and constructs the personalized reward model $r(\cdot;\theta)$ **on the fly**. This makes T-POP applicable in scenarios where traditional methods would fail to initiate.
> >
> > **2. Dynamic Calibration to Preference Intensity via Minimal Feedback**
> > Unlike static methods that freeze the reward function during inference, T-POP establishes an **online closed-loop system** that continuously calibrates the model to the specific **intensity and granularity** of user preferences.
> > * **Feedback Efficiency:** T-POP achieves this using only lightweight **binary** preference feedback, which imposes significantly lower cognitive load than methods requiring explicit scoring or detailed instructions.
> > * **Adaptive Refinement:** Through the iterative updates of the reward model , T-POP can detect not just *what* the user prefers, but *how strongly* the generation should be steered. As shown in our experiments (Fig. 2), the model rapidly adjusts the reward scores to converge towards the user's desired level of personalization . This dynamic self-correction prevents the model from being stuck in a suboptimal state (e.g., under-steering or reward hacking), a common pitfall for static decoding-time control methods.
> >
> > ---
> >
> > Thank you again for your thoughtful comments. We hope our additional results and clarifications will improve your evaluation of our work.

---

> ### Author Response · Authors · 2025-11-27
> **Follow-up on rebuttal**
>
> Dear Reviewer gvEW,
>
> We would like to thank you for your constructive feedback and the effort put into reviewing our work.
>
> This is a gentle reminder that our rebuttal is available for your review. With less than a week remaining in the discussion period, we would appreciate it if you could take a moment to review our rebuttal.
>
> Please let us know if you have further questions; we would be happy to provide additional clarifications.
>
> Best,
> Authors

---

### Author Response · Authors · 2025-11-30
**Summary of Revisions**

We sincerely thank the reviewers and the Area Chair for their constructive feedback and the time dedicated to evaluating our work. We are encouraged by the recognition of our "novel idea" (Reviewer gvEW), "well-motivated" problem (Reviewer wJsq), and "inspiring" approach (Reviewer gN7W).

We have addressed all reviewers' concerns in our rebuttal and revised our paper accordingly. All major revisions in the attached PDF are highlighted in **blue**.

### Summary of Revisions and Responses

* **Concerns regarding inference latency and computational overhead.**
    * **Reviewers:** gvEW, gN7W
    * **Response & Revision:** We acknowledged the latency concern and introduced a "Continuous Deployment" strategy via **Asynchronous Online Updates**. This decouples model training from user interaction, effectively reducing perceived training latency to zero. We also provided a detailed wall-clock time analysis comparing T-POP against the SOTA baseline (AMULET), showing our token-level latency (0.18s) is within the same order of magnitude as AMULET (0.09s).
    * **Revision:** We added the description of the asynchronous update strategy in **Section 3.2** and the wall-clock latency analysis in **Section 5 (Table 5)** and **Appendix C.3**.

* **Questions regarding data efficiency and the "Cold-Start" performance (Is 20 turns too burdensome?).**
    * **Reviewers:** gvEW, wJsq, HUPh
    * **Response & Revision:** We clarified that effective personalization does not require waiting for convergence. We added a new experiment analyzing performance at very early stages ($T=5$ and $T=10$). Results show T-POP achieves an ArmoRM score of **0.56** after just **5 interactions**, significantly outperforming the Base model (0.39) and strong baselines like Linear Alignment (0.53). This demonstrates immediate utility for the user.
    * **Revision:** We included the early-stage performance analysis in **Appendix C.1 (Table 7)** and discussed these findings in **Section 4.2**.

* **Request for theoretical guarantees for the dueling bandit and token-level approach.**
    * **Reviewers:** wJsq, HUPh, gN7W
    * **Response & Revision:** We explicitly connected our method to established theoretical bounds. We cited **Verma et al. (2024)** for the regret bounds of Neural Dueling Bandits ($\tilde{O}(d_{eff}\sqrt{T})$) regarding preference learning, and **Shin et al. (2025)** for Tokenized Bandits ($\tilde{O}(L\sqrt{T})$) regarding sequential decoding under the DDMC assumption. This provides a rigorous foundation for our algorithmic design.
    * **Revision:** We added the theoretical guarantees and discussion at the end of **Section 3.2**.

* **Justification for the specific uncertainty sampling strategy (Ablation Study).**
    * **Reviewers:** HUPh, gN7W
    * **Response & Revision:** We conducted a new ablation study comparing our gradient-based uncertainty metric against Entropy-based, Boltzmann (temperature sampling), and Random exploration strategies. Results confirm that our method, which captures **epistemic uncertainty** (parameter uncertainty), significantly outperforms heuristics that only capture aleatoric uncertainty.
    * **Revision:** We added the exploration strategy ablation study in **Section 5 (Table 4)** and **Appendix C.2**.

* **Concerns about scalability, memory usage, and storage per user.**
    * **Reviewers:** wJsq
    * **Response & Revision:** We clarified that the Reward Model is a lightweight MLP (~16MB), negligible compared to the LLM backbone. To address memory concerns regarding the covariance matrix, we explicitly stated our use of **Diagonal Approximation**, a standard technique that reduces memory complexity from quadratic to linear ($\mathcal{O}(d)$), ensuring scalability for large user bases.
    * **Revision:** We clarified the Diagonal Approximation and model architecture details in **Appendix B**.

* **Questions regarding the fairness of comparison with static baselines (LA, AMULET) and novelty.**
    * **Reviewers:** gvEW, HUPh, gN7W
    * **Response & Revision:** We emphasized that T-POP addresses the **absolute cold-start** problem where static baselines fail due to a lack of prior data. The requirement for online feedback is a feature, not a flaw, enabling zero-shot personalization. We also clarified the distinction from AMULET: T-POP uses comparative feedback (easier for users than explicit text instructions) and a different algorithmic foundation (Dueling Bandits vs. FTRL).
    * **Revision:** We strengthened the discussion on problem setting and novelty in **Section 1** and **Section 6**.

---

### Meta-Review · Area_Chair_CMJf · 2025-12-26

**Summary:**

Reviewers viewed the problem as timely and the high-level idea as interesting, but their concerns collectively supported a reject recommendation. The primary concern was deployability: the method depends on repeated online pairwise comparisons, which can impose meaningful cognitive load because users must read and compare two answers each round, and it introduces additional compute/latency and per-user state that may be hard to scale. Reviewers also questioned novelty and positioning, noting similarities to prior test-time alignment and online decoding realignment (e.g., AMULET) and asking for clearer evidence that the dueling-bandit uncertainty component provides a distinct advantage rather than an incremental modification. Finally, reviewers raised soundness and evaluation issues, including limited paper-specific theory beyond citing existing results and requests for stronger ablations and broader baseline coverage to substantiate the claimed benefits.

**Reviewer Concerns:**

The rebuttal addressed feasibility concerns partially. The wall-clock breakdown and the asynchronous update strategy help, but they do not eliminate the practical cost because T-POP still adds nontrivial end-to-end latency and extra computation compared to non-interactive baselines (gvEW, gN7W). The rebuttal also strengthens the empirical case with early-iteration results and an ablation supporting the chosen uncertainty/exploration design (gvEW, wJsq, HUPh, gN7W).

However, major concerns remain outstanding. Pairwise feedback is “binary” but still demanding because users must read and compare two answers repeatedly, and the rebuttal does not quantify this cognitive/time burden or show an interaction-sparing protocol (gvEW, HUPh, gN7W). Novelty and soundness skepticism relative to prior test-time alignment and online decoding realignment (e.g., AMULET) may also persist, since the added theory largely cites existing results and may not fully justify the specific token-level bandit/uncertainty design end-to-end (HUPh, gN7W).

**Reviewer Scores:**

- gvEW (4 → 4 or 5): The added wall-clock analysis + asynchronous updates and the early-iteration results would likely alleviate their biggest practicality worries, so they could bump to a weak accept; but remaining concerns about user burden and broader baseline coverage could keep them at 4.

 - wJsq (6 → 6 or 7): The rebuttal directly addresses their main critique (per-user storage / covariance scalability) and adds theory pointers; I’d expect them to stay accept and possibly increase slightly if they find the diagonal approximation explanation convincing.

 - HUPh (2 → 2 or 3): The exploration ablation and added theory connections help, but their core objections were about incremental novelty vs AMULET and the heuristic feel of token-level uncertainty; I’d expect at most a modest move to a weak reject (3), and it could remain 2.

 - gN7W (2 → 3): Their main asks were wall-clock latency accounting and a core sampling-policy ablation; the rebuttal provides both, so they’d likely soften from strong reject to weak reject, but baseline fairness/interaction-assumption mismatch could still prevent an accept

---

### Decision · Program_Chairs · 2026-01-26

Reject